# Mechanical instability and interfacial energy drive biofilm morphogenesis

Jing Yan[1,2†], Chenyi Fei[2†], Sheng Mao[1†], Alexis Moreau[1], Ned S Wingreen[2], Andrej Košmrlj[1], Howard A Stone[1*], Bonnie L Bassler[2,3*]

[1]Department of Mechanical and Aerospace Engineering, Princeton University, Princeton, United States; [2]Department of Molecular Biology, Princeton University, Princeton, United States; [3]The Howard Hughes Medical Institute, Chevy Chase, United States

**Abstract** Surface-attached bacterial communities called biofilms display a diversity of morphologies. Although structural and regulatory components required for biofilm formation are known, it is not understood how these essential constituents promote biofilm surface morphology. Here, using *Vibrio cholerae* as our model system, we combine mechanical measurements, theory and simulation, quantitative image analyses, surface energy characterizations, and mutagenesis to show that mechanical instabilities, including wrinkling and delamination, underlie the morphogenesis program of growing biofilms. We also identify interfacial energy as a key driving force for mechanomorphogenesis because it dictates the generation of new and the annihilation of existing interfaces. Finally, we discover feedback between mechanomorphogenesis and biofilm expansion, which shapes the overall biofilm contour. The morphogenesis principles that we discover in bacterial biofilms, which rely on mechanical instabilities and interfacial energies, should be generally applicable to morphogenesis processes in tissues in higher organisms.
DOI: https://doi.org/10.7554/eLife.43920.001

**\*For correspondence:**
hastone@princeton.edu (HAS);
bbassler@princeton.edu (BLB)

[†]These authors contributed equally to this work

**Competing interests:** The authors declare that no competing interests exist.

## Introduction

Many of the stunning morphologies that distinguish living entities do not arise exclusively from gene expression programs, but rather from overarching contributions from mechanical forces (*Heisenberg and Bellaïche, 2013*; *Thompson, 1992*; *Yamada and Cukierman, 2007*). Such morphomechanical processes include the formation of ripple-shaped leaves (*Liang and Mahadevan, 2009*), tendrils and flowers (*Gerbode et al., 2012*; *Liang and Mahadevan, 2011*), as well as the dorsal closure and apical constriction-mediated epithelial folding processes that take place during *Drosophila* embryonic development (*He et al., 2014*; *Solon et al., 2009*). One key feature is common to many of these morphogenic transformations: two or more layers of biomaterials are attached to one another but each grows at a different rate (*Wang and Zhao, 2015*). Inevitably, such growth mismatches generate mechanical stresses, and corresponding shape instabilities, which depend on the mechanical and other material properties of the biological constituents, as well as their geometries. Some examples include villi formation during the development of the human gut and formation of gyri and sulci during cerebrum development (*Shyer et al., 2013*; *Budday et al., 2015*; *Tallinen et al., 2016*).

Though ancient in their evolutionary origin, bacterial cells can also display intricate developmental patterns, particularly when they exist in the community lifestyle known as biofilms (*Hobley et al., 2015*; *Humphries et al., 2017*; *Persat et al., 2015*). Biofilms are surface-associated bacterial communities that are embedded in a polymer matrix (*O'Toole et al., 2000*; *Thongsomboon et al., 2018*) and are a predominant growth mode for bacteria in nature (*Hall-Stoodley et al., 2004*; *Humphries et al., 2017*). Biofilms can be beneficial, for example in waste-water treatment

**eLife digest** Engineers have long studied how mechanical instabilities cause patterns to form in inanimate materials, and recently more attention has been given to how such forces affect biological systems. For example, stresses can build up within a tissue if one layer grows faster than an adjacent layer. The tissue can release this stress by wrinkling, folding or creasing.

Though ancient and single-celled, bacteria can also develop spectacular patterns when they exist in the lifestyle known as a biofilm: a community of cells adhered to a surface. But do mechanical instabilities drive the patterns seen in biofilms?

To investigate, Yan, Fei, Mao et al. grew biofilms of the bacterium called *Vibrio cholerae* – which causes the disease cholera – on solid, non-growing 'substrates'. This work revealed that as the biofilms grow, their expansion is constrained by the substrate, and this situation generates mechanical stresses. To release the stresses, the biofilm initially folds to form wrinkles. Later, as the biofilm expands further, small parts of it detach from the substrate to form blisters. The same forces that keep water droplets spherical (known as interfacial forces) dictate how the blisters evolve, interact, and eventually shape the expanding biofilm. Using these principles, Yan et al. could engineer the biofilm into desired shapes.

Collectively, the results presented by Yan et al. connect the shape of the biofilm surface with its material properties, in particular its stiffness. Understanding this relationship could help researchers to develop new ways to remove harmful biofilms, such as those that cause disease or that damage underwater structures. The stiffness of biofilms is already known to affect how well bacteria can resist antibiotics. Future studies could look for new genes or compounds that change the material properties of a biofilm, thereby altering the biofilm surface.

DOI: https://doi.org/10.7554/eLife.43920.002

(*Nerenberg, 2016*), but they also cause significant problems in health and industry (*Costerton et al., 1999*; *Drescher et al., 2013*) because they are resistant to physical perturbations and to antibiotics (*Kovach et al., 2017*; *Meylan et al., 2018*). Biofilms on surfaces undergo morphogenic transformations, beginning as smooth colonies and, over time, developing complex morphological features (*Beyhan and Yildiz, 2007*). Genes specifying matrix components that enable polysaccharide production, cell-surface adhesion, and cell–cell adhesion are required for the morphological transition (*Hobley et al., 2015*). However, the underlying mechanisms that dictate how these biofilm matrix components direct overall morphology are not well-understood. One model focuses on the differential spatial regulation of genes encoding matrix components as the key driver of biofilm morphogenesis (*Okegbe et al., 2014*). Another model suggests that localized cell death serves as an outlet for mechanical stresses and thus determines biofilm morphology (*Asally et al., 2012*). Most recently, theory has been put forward to suggest the possibility that global mechanical instabilities are involved in the development of biofilm morphology (*Zhang et al., 2016*; *Zhang et al., 2017*).

Here, by combining quantitative imaging, biomaterial characterization, mutant analyses, and mechanical theory, we show that the mismatch between the growing biofilm layer and the non-growing substrate causes mechanical instabilities that enable the biofilm to transition from a flat to a wrinkled film, and subsequently to a partially detached film containing delaminated blisters. The sequential instabilities that the film undergoes, coupled with the generation and annihilation of interfaces, drive the evolution of biofilm topography. Our results demonstrate that bacterial biofilms provide a uniquely tractable system for the quantitative investigation of mechanomorphogenesis.

## Results

### A mechanical instability model for biofilm morphogenesis

Our central hypothesis is that biofilm morphogenesis is driven by mechanical instabilities that arise from the growth mismatch between an expanding biofilm and the non-growing substrate to which it adheres. To garner evidence for this idea, we grew biofilms on agar plates, which enabled us to control the mechanical properties of the substrate by changing the agar concentration (*Nayar et al.,*

*2012*). We employed a commonly used *Vibrio cholerae* strain that lacks motility and constitutively produces biofilms (*Beyhan and Yildiz, 2007*; *Yan et al., 2017*). This strain (denoted WT in the present work) produces biofilms that have disordered cores decorated with radial features extending to the rims (*Figure 1A*). Indeed, biofilm surface morphology changes with increasing agar concentration: the spacing between the peripheral, radial features is reduced and their amplitudes become more homogeneous (*Figure 1—figure supplement 1*).

Encouraged by the observations described above and inspired by models developed to describe mechanical instabilities in abiotic materials systems (*Li et al., 2012*), here we propose a mechano-morphogenesis model for biofilms (*Figure 1B*). The biofilm originates as a flat film. Its volume increases over time due to cell proliferation and matrix production. If the biofilm were not attached

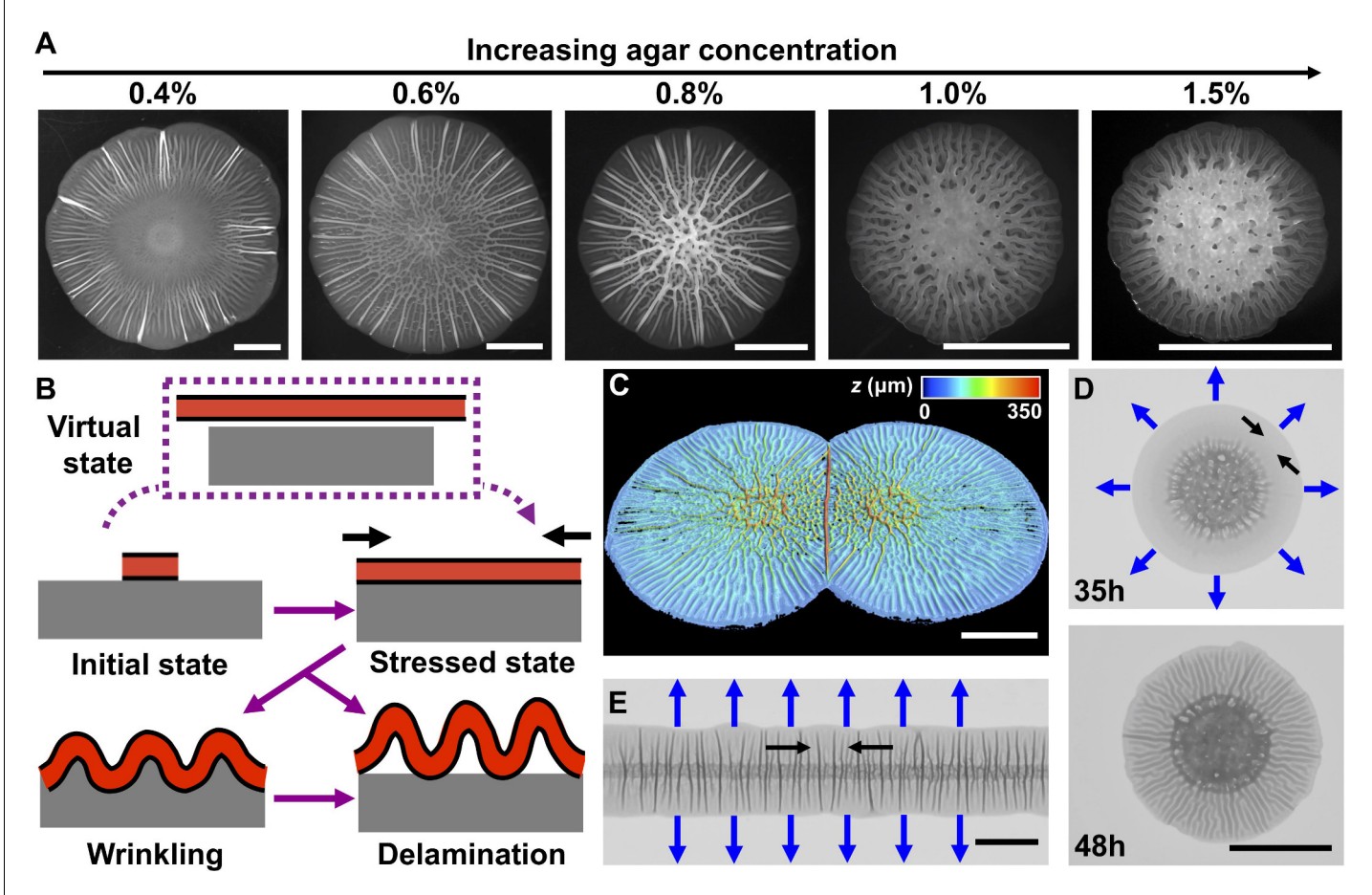

**Figure 1.** Mechanical instability drives *V. cholerae* biofilm morphogenesis. (A) Bright-field images of biofilms grown for 2 days on the designated percentages of agar. (B) Schematic of the wrinkling and delamination processes that occur during biofilm expansion. Red with a black outline denotes the biofilm. Gray denotes the substrate, agar in this case. (C) Three-dimensional (3D) profile of two colliding biofilms, initially inoculated 9 mm apart, grown on a 0.6% agar plate for 36 hr. (D) Transmission image of a *V. cholerae* biofilm grown for 35 hr (*top*) and 48 hr (*bottom*) on a 1.0% agar plate. (E) Transmission image of a *V. cholerae* biofilm inoculated as a line and grown for 30 hr on a 0.5% agar plate. In panels (D) and (E), blue arrows denote the expansion directions, and black arrows denote the tangential directions along which compressive stress accumulates. All scale bars are 5 mm.

DOI: https://doi.org/10.7554/eLife.43920.003

The following source data and figure supplements are available for figure 1:

**Figure supplement 1.** Quantification of *V. cholerae* biofilm surface morphologies.
DOI: https://doi.org/10.7554/eLife.43920.004

**Figure supplement 1—source data 1.** Quantitation of *V. cholerae* biofilm surface morphologies.
DOI: https://doi.org/10.7554/eLife.43920.005

**Figure supplement 2.** Wrinkling and delamination transitions are rapid.
DOI: https://doi.org/10.7554/eLife.43920.006

to a substrate, it would grow into a stress-free state to cover a large area (*Figure 1B*, top, 'virtual state'). However, the non-expanding agar substrate constrains biofilm expansion. Thus, biofilms are always subject to compressive stress (*Figure 1B*, middle right), which we hypothesize drives the surface morphology. Indeed, a biofilm growing at an air–liquid interface, not limited or compressed by a substrate, exhibits no surface features (*Video 1*).

According to mechanical instability theories, surface-adhered films under compression have several pathways to release compressive stress (*Wang and Zhao, 2015*). For example, the film can buckle out of the growth plane and deform together with the substrate into a periodically wrinkled pattern (*Figure 1B*, bottom left). In this mode, the compressive stress is released by film bending and substrate deformation. Alternatively, the film can directly delaminate from the substrate to form 'blisters' (*Figure 1B*, bottom right) (*Vella et al., 2009*), leaving the substrate essentially undeformed. An extra interfacial energy penalty is paid for delamination because new interfaces are generated, so direct delamination occurs in systems with film–substrate adhesion energies that are much smaller than their elastic deformation energies. Biofilms possess finite adhesion strength (~ 5 mJ/m$^2$), which is the same order of magnitude as the deformation energy of the soft substrate (*Yan et al., 2018*). Thus, we suggest that biofilms could first wrinkle, and subsequently delaminate as growth gradually builds up compressive stress (*Figure 1—figure supplement 2*). According to this mechanomorphogenesis model, we should be able to change the biofilm topography by changing the spatial distribution of the mechanical stress. To this end, we inoculated two *V. cholerae* biofilms onto the same agar plate and allowed them to collide. Indeed, a large localized blister formed at the collision front where mechanical stress is most concentrated (*Figure 1C*; *Video 2*).

Our mechanomorphogenesis model provides an intuitive explanation for the commonly observed biofilm surface pattern of a disordered core surrounded by radial features at the edge (*DePas et al., 2013*; *Okegbe et al., 2014*; *Wilking et al., 2013*). Soon after the initial expansion of the biofilm, growth occurs primarily at the edge of the biofilm because of nutrient limitation at the center of the biofilm (*Liu et al., 2015*; *Yan et al., 2017*; and *Figure 1—figure supplement 2*). At the biofilm center, cell death has been shown to drive pattern formation (*Asally et al., 2012*). However, in the biofilm periphery, which is the region of focus of the current study, wrinkling and delamination occur with no preceding localized cell death (*Figure 1—figure supplement 2*). In this outer region, mechanical instabilities dominate the pattern formation and its wavelength. Directionality at the edge stems from the asymmetry between radial and tangential compressive stresses on the expanding front (*Figure 1D*). During cell proliferation, radial compressive stress is partially relieved by new biomass extending the biofilm boundary (*Zhang et al., 2016*). By contrast, in the tangential direction, compressive stress becomes concentrated because there is no analogous relaxation mechanism. Therefore, starting from a flat film, a growing biofilm will undergo mechanical instabilities predominantly in the tangential direction, leading to radial wrinkling, and later, to delamination patterns (*Figure 1D*). By contrast, in the interior region of a biofilm, compressive stress occurs in both the radial and tangential directions, giving rise to a network containing both radially and tangentially oriented features (*Figure 1A,D*). To demonstrate that pattern directionality is determined by expansion anisotropy, we changed the biofilm growth geometry by inoculating cells starting from a line so that the biofilm would extend quasi-unidirectionally (*Video 2*). In this geometry, compressive stress along the inoculation line is higher than that perpendicular to the line (the expanding direction). Therefore, wrinkles or blisters occur perpendicular to the biofilm line (*Figure 1E*).

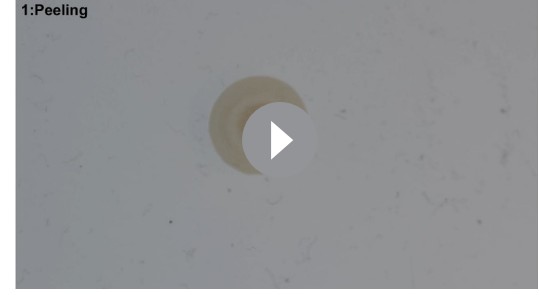

**Video 1.** Part 1: A *V. cholerae* biofilm grown for 24 hr on 0.6% agar medium was peeled off of the substrate by the capillary method using LB medium as the liquid starting from the bottom left. The movie is played in real time. Part 2: The peeled biofilm from Part 1 grew at the air–liquid interface over time. Imaging began immediately after peeling and its total duration is 6 hr with 5-min time steps. The field of view is 73.0 mm × 48.3 mm.

DOI: https://doi.org/10.7554/eLife.43920.007

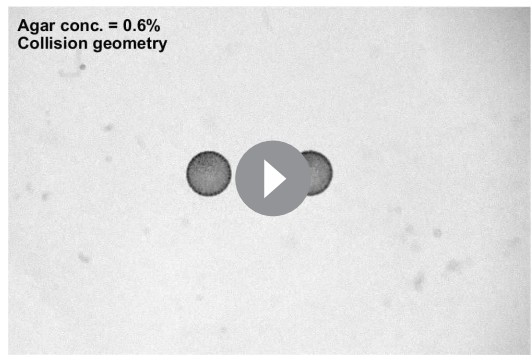

**Video 2.** Part 1: Collision of two *V. cholerae* biofilms grown on medium containing 0.6% agar. Imaging began 5 hr after inoculation and has a total duration of 75 hr with 15 min time steps. Biofilms were separated by 9 mm at the time of inoculation. At *t* = 20 hr, the biofilms begin to contact one another. The additional compressive stress present at the collision front leads to the formation of a large blister in the middle. The field of view is 41.5 mm × 27.7 mm. Part 2: Growth of a *V. cholerae* biofilm on medium containing 0.5% agar after cells were inoculated in a line. Imaging began 5 hr after inoculation and has a total duration of 72 hr with 15 min time steps. The field of view is 50.2 mm × 33.3 mm.

DOI: https://doi.org/10.7554/eLife.43920.008

# A trilayer mechanical model predicts the biofilm wrinkling wavelength

Mechanical instability theory predicts that, for a film–substrate system that is subject to compressive stress, the wrinkling wavelength is determined exclusively by the thickness and mechanical properties of the relevant materials (*Huang et al., 2005*). If so, we would expect the wrinkle wavelength to change with the mechanical properties of the biofilm and substrate but to be independent of the growth stage and geometry of the biofilm. To extract the wrinkle wavelength, we imaged the biofilm morphogenesis process over 72 hr and quantified the periodicity of radial stripes (*Figure 2—figure supplement 1*; *Videos 3–5*). We note that blisters emerge from wrinkles and that they inherit the wavelength of wrinkles, so we do not distinguish between the two in this analysis. We quantified the number of wrinkles or blisters $N$ as a function of radial distance $r$ from the biofilm center at different times. We found a linear relationship between $N$ and $r$ (*Figure 2A*, *Figure 2—figure supplement 1*). The slope has a geometrical origin: $N = (2\pi/\lambda)r$ in which $\lambda$ is the inherent wavelength of the system irrespective of the time in the developmental process or the location in the overall pattern (except at the biofilm core). A constant wavelength $\lambda$ also means that radial wrinkles or blisters must bifurcate to maintain constant spacing as $r$ increases, and indeed, we observed this to be the case (*Figure 2A*, inset). We also confirmed that the same $\lambda$ was maintained when cells were inoculated in the line geometry and grew quasi-unidirectionally (*Figure 2—figure supplement 1*). We conclude that the wavelength of wrinkles or blisters reflects an intrinsic physical property of the biomechanical system.

Mechanical instability theory also predicts how the wavelength varies with the stiffness contrast between the biofilm and the substrate. Classical linear stability analysis for bilayer film–substrate systems

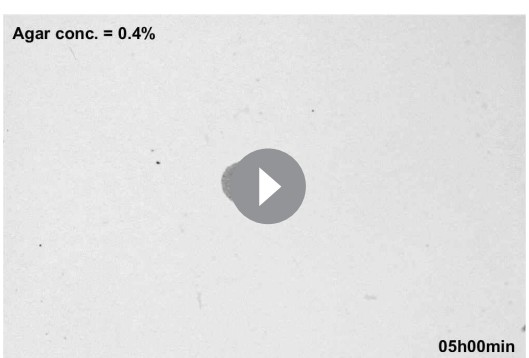

**Video 3.** Growth of a *V. cholerae* biofilm on medium containing 0.4% agar. Imaging began 5 hr after inoculation and has a total duration of 75 hr with 15 min time steps. The field of view is 41.5 mm × 27.7 mm.

DOI: https://doi.org/10.7554/eLife.43920.019

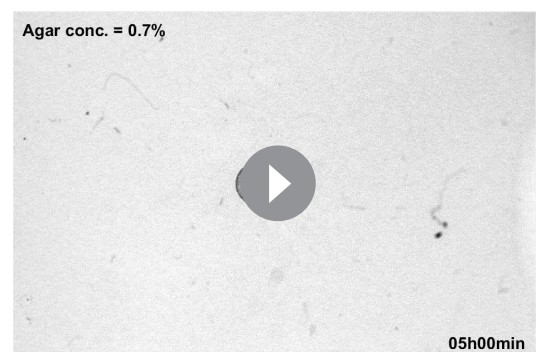

**Video 4.** Growth of a *V. cholerae* biofilm on medium containing 0.7% agar. Imaging began 5 hr after inoculation and has a total duration of 75 hr with 15 min time steps. The field of view is 41.5 mm × 27.7 mm.

DOI: https://doi.org/10.7554/eLife.43920.020

predicts that $\lambda$, normalized by the film thickness $h$, should be equal to $2\pi(G_f/3G_s)^{1/3}$, in which $G_f$ and $G_s$ are the shear modulus of the film and the substrate, respectively (*Chen and Hutchinson, 2004*; *Huang et al., 2005*). The 1/3 power law is a result of the competition between the energy cost to deform the film and that to deform the substrate. To test whether this relationship applies to biofilms, we measured $\lambda$, $h$, $G_s$, and $G_f$ for all growth conditions. $G_f$ varies minimally over a wide range of agar concentrations, whereas $G_s$ varies by almost three orders of magnitude for agar concentrations from 0.4% to 3% (*Supplementary file 1* Table S1). Plotting $\lambda/h$ versus $G_f/G_s$ on a log-log scale (*Figure 2B*) reveals the characteristic scaling power law of 1/3, indicating the applicability of mechanical instability theory to biofilm morphogenesis.

One key discrepancy exists between the experimental measurements and the bilayer model. Bilayer theory predicts that, if $G_f/G_s < 1.3$, the substrate is too stiff for the flat-to-wrinkling transition to occur (*Wang and Zhao, 2015*). However, wrinkling occurs in our experiments for $G_f/G_s$ well below 1.3, corresponding to agar concentrations $\geq$ 0.7%. To reconcile this discrepancy, we considered that a third soft, intermediate layer could exist between the growing biofilm and the non-growing substrate, which has been shown to allow wrinkling behavior even at low $G_f/G_s$ ratios (*Lejeune et al., 2016a*).

To acquire evidence for an intermediate layer, we employed a capillary peeling method in which biofilms are gently dipped into water and the capillary force peels the biofilm off the substrate without destroying the biofilm or the underlying surface (*Figure 2—figure supplement 2*) (*Yan et al., 2018*). Prior to peeling, using reflective confocal microscopy, the total biofilm thickness $h$ was measured. After peeling, a residual layer remained on the substrate with a thickness $h_r$ (*Figure 2C*). Our preliminary analysis suggests that this layer consists primarily of matrix polysaccharide (*Figure 2—figure supplements 2* and *3*). Thus, the corrected biofilm thickness $h_f$ was obtained as $h_f = h - h_r$. We replotted our data using $h_f$ (*Figure 2D*, *Figure 2—figure supplement 4*). To rationalize the replotted curve, we took advantage of recent modeling efforts concerning multi-layer wrinkling phenomena (*Lejeune et al., 2016a*). The only unknown parameter in our work is the shear modulus of the residual layer, $G_r$. In our theoretical model, we use a residual layer thickness $h_r = 0.3h_f$, which was obtained from our experimental measurements, and we left $G_r/G_f$ as a fitting parameter (*Figure 2—figure supplement 4*). The trilayer model qualitatively and quantitatively captures our experimental observations. Qualitatively, with a soft intermediate layer, the wrinkling pattern persists even when the substrate becomes stiffer than the biofilm ($G_s > G_f$). Unlike the bilayer model, in which the substrate is deformed by the wrinkling film, in the trilayer model, the soft interfacial layer assumes the major share of the deformation, effectively reducing the substrate stiffness (*Figure 2D*, *Figure 2—figure supplement 4*) (*Lejeune et al., 2016a*). Quantitatively, predictions from the trilayer model recapitulate the prominent features of the revised plot: $\lambda/h_f$ scales according to the bilayer model as $2\pi(G_f/G_s)^{1/3}$ for large $G_f/G_s$ ratios, but increasingly deviates from the 1/3 scaling law for smaller $G_f/G_s$ values. In the low $G_f/G_s$ regime, wrinkling is increasingly controlled by the soft intermediate layer. An intermediate layer stiffness of $G_r = 0.1G_f$ allows the trilayer model to best fit our experimental data over all conditions.

## The biofilm wrinkling-to-delamination transition is controlled by interfacial energy and substrate stiffness

We next investigated the second transition predicted by our mechanomorphogenesis model: wrinkling-to-delamination. Whether and when a film–substrate system undergoes delamination is controlled by a competition between the adhesion energy between layers, $\Gamma$, and the elastic energy in the substrate. A dimensionless term $\Gamma^*$, defined as $\Gamma/(h_f G'_s)$ in which $G'_s$ is the effective substrate modulus taking into account the residual layer (*Lejeune et al., 2016a*; see also Materials and

Agar conc. = 1.0%

05h00min

**Video 5.** Growth of a *V. cholerae* biofilm on medium containing 1.0% agar. Imaging began 5 hr after inoculation and has a total duration of 72 hr with 15 min time steps. The field of view is 24.0 mm × 16.0 mm.
DOI: https://doi.org/10.7554/eLife.43920.021

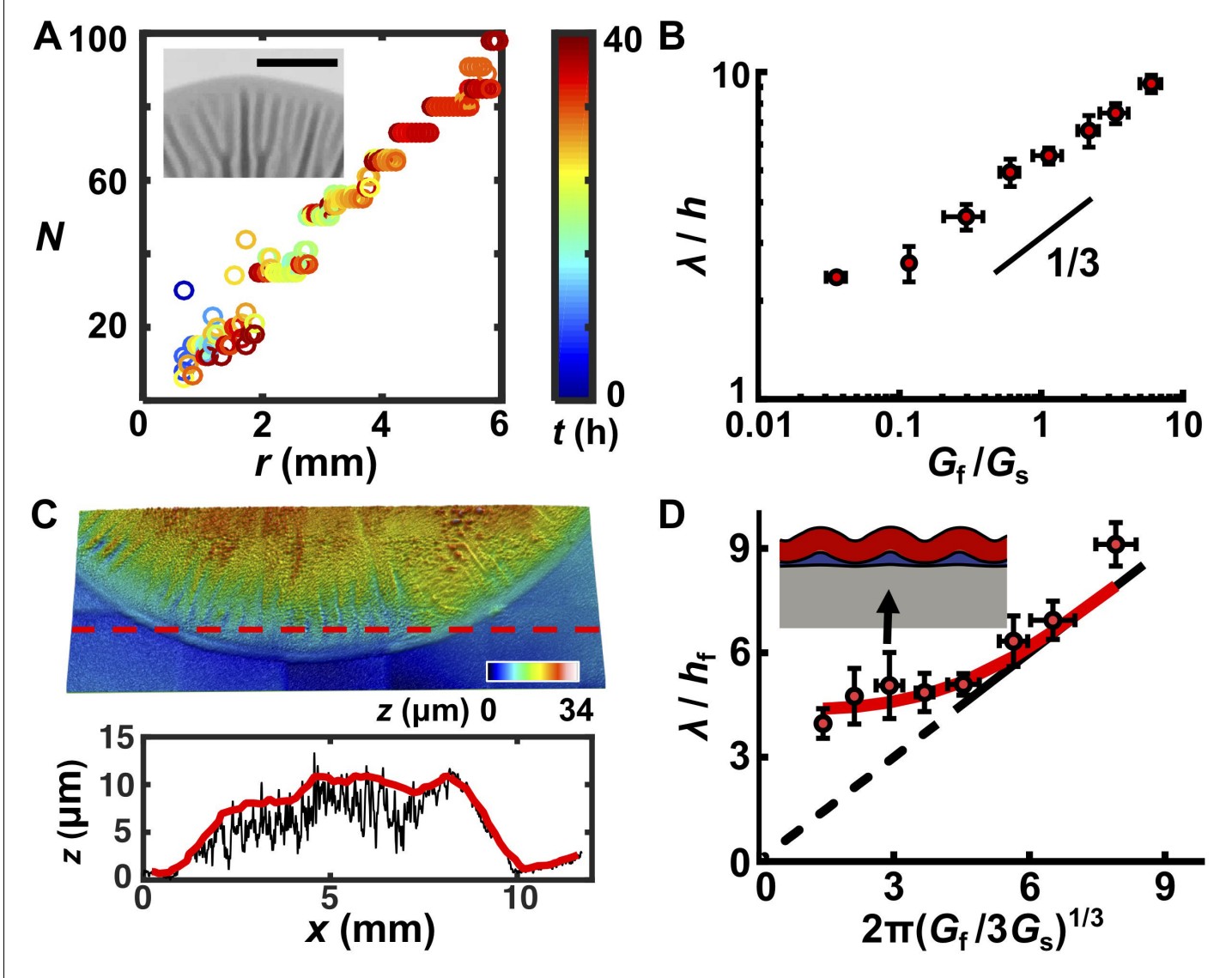

**Figure 2.** A trilayer mechanical model predicts the intrinsic wavelength of the biofilm pattern. (**A**) Number of wrinkles or blisters $N$ versus the radial coordinate $r$ during biofilm growth. The color scale indicates growth time $t$. Inset: closeup transmission image of a growing biofilm showing that wrinkles or blisters bifurcate to maintain a constant $\lambda$. Agar concentration: 0.7%, scale bar: 2 mm. (**B**) The scaling relationship between $\lambda$ (normalized by the biofilm thickness $h$) and the shear modulus ratio $G_f/G_s$ between the biofilm and the agar substrate. The black line indicates a slope of 1/3 on a log-log scale. (**C**) Characterization of the residual layer. *Top*: 3D topography of the residual layer after peeling a biofilm off of an agar substrate. *Bottom*: height profile extracted along the contour indicated by the dashed red line in the *top* panel. Both the raw (black) and smoothed (red) data, from which the residual layer thickness $h_r$ was calculated, are shown. Agar concentration: 0.5%. (**D**) Replot of the data in panel (**B**) taking into account the residual layer. The corrected biofilm thickness $h_f$ was obtained by subtracting the residual thickness $h_r$ from the total thickness $h$. The solid portion of the black line corresponds to the prediction from the bilayer model, which applies only to x coordinates greater than 4.75 (*Wang and Zhao, 2015*). The dashed portion of the black line is an extrapolation to zero from the bilayer prediction provided as a guide to the eye. The red line is the fitted data from the trilayer model in which the stiffness contrast between the residual and biofilm layers $G_r/G_f$ is treated as a fitting parameter while holding $h_r/h_f = 0.3$. Inset: finite-element simulation of the trilayer model undergoing wrinkling instability. Red denotes the biofilm. Gray denotes the substrate. Blue denotes the residual layer. Simulation parameters were chosen to mimic the growth condition on 1.0% agar (black arrow). Data are represented as mean ± std with $n = 3$.

DOI: https://doi.org/10.7554/eLife.43920.009

The following source data and figure supplements are available for figure 2:

**Source data 1.** Experimental measuremants of biofilm residual layer thicknesses and wavelengths and predictions from trilayer wrinkling theory.
DOI: https://doi.org/10.7554/eLife.43920.010

**Figure supplement 1.** Analysis of intrinsic wavelengths in the morphologies of biofilms.

*Figure 2 continued*

DOI: https://doi.org/10.7554/eLife.43920.011

**Figure supplement 1—source data 1.** Biofilm wavelength analysis.

DOI: https://doi.org/10.7554/eLife.43920.012

**Figure supplement 2.** Capillary peeling reveals a residual layer between the biofilm and the substrate.

DOI: https://doi.org/10.7554/eLife.43920.013

**Figure supplement 2—source data 2.** Thicknesses of the biofilm and residual layers.

DOI: https://doi.org/10.7554/eLife.43920.014

**Figure supplement 3.** The biofilm residual layer consists primarily of polysaccharide.

DOI: https://doi.org/10.7554/eLife.43920.015

**Figure supplement 3—source data 3.** Cell counts in biofilm and residual layers.

DOI: https://doi.org/10.7554/eLife.43920.016

**Figure supplement 4.** The trilayer biofilm morphology model predicts the wrinkling wavelength observed in the experiments.

DOI: https://doi.org/10.7554/eLife.43920.017

**Figure supplement 4—source data 4.** Theoretical and computational models for trilayer wrinkling.

DOI: https://doi.org/10.7554/eLife.43920.018

methods), was used previously to quantify the relative importance of the two energies (*Wang and Zhao, 2015*). We recently measured the biofilm–agar interfacial adhesion energy $\Gamma \sim 5$–$10$ mJ/m$^2$ (*Yan et al., 2018*). Hence, $\Gamma^*$ is in the order of 0.01–1 in the current system, making delamination highly likely to occur during biofilm growth. In the context of the trilayer model, delamination takes place at the weakest interface, which is between the biofilm and the residual layer.

To access the wrinkling-to-delamination transition experimentally, we simultaneously imaged the growing biofilm from the top and the side (*Figure 3A*, *Figure 2—figure supplement 1*). Radial wrinkles developed into blisters when growth proceeded beyond ~ 36 hr. At low agar concentrations, large amplitude blisters emerged among small amplitude wrinkles (*Figure 3A*). At higher agar concentrations, additional wrinkles developed into blisters, although with amplitudes smaller than those on low concentration agar substrates. We verified these findings using optical profiling to capture the full three-dimensional (3D) height information of the entire biofilm (*Figure 3B*). To peer inside blisters, we imaged cross-sectioned biofilms grown from cells producing fluorescence from *mKate2* (*Figure 3C*). At low agar concentration (i.e., 0.6%), only a small fraction of wrinkles were detached from the substrate in the form of blisters (*Figure 3—figure supplement 1*). By contrast, at high agar concentration (i.e. 1.0%), nearly all wrinkles had developed into blisters. In the cross-sectional images, voids were clearly present underneath the blisters, which were presumably filled with liquid (*Wilking et al., 2013*). *Figure 3D* quantifies the positive correlation between the percentage of wrinkles that converted to blisters at the biofilm edge and the substrate agar concentration.

To rationalize the dependence of the delamination pattern on agar concentration, it is useful to recall the notion of normalized adhesion energy, $\Gamma^*$. On stiff substrates, $\Gamma^*$ is small, so delamination is favored over wrinkling. Blisters form extensively but they are small because they share the overall compression. On soft substrates, $\Gamma^*$ is large, so blisters form only infrequently while the majority of the biofilm remains attached to the substrate. In this case, the isolated blisters concentrate the compressive strain and become larger than those on a stiff substrate. We hypothesized that the locations of blisters on soft substrates are defined by surface defects that trigger local delamination. This hypothesis is consistent with the observed heterogeneous sizes of blisters in biofilms grown on soft substrates. Specifically, we argue that blisters emerge at different times and at different locations in growing biofilms depending on when a surface defect is encountered during biofilm expansion. The different ages of blisters naturally lead to their heterogeneous heights. To test this possibility, we made surface imperfections in the soft agar substrate at defined positions (*Figure 3E*). Indeed, these imperfections dictated the exact locations at which blisters formed as the biofilm expanded. By contrast, on stiff substrates, delamination occurred along the entire biofilm rim, irrespective of the predefined surface imperfections (*Figure 3E*).

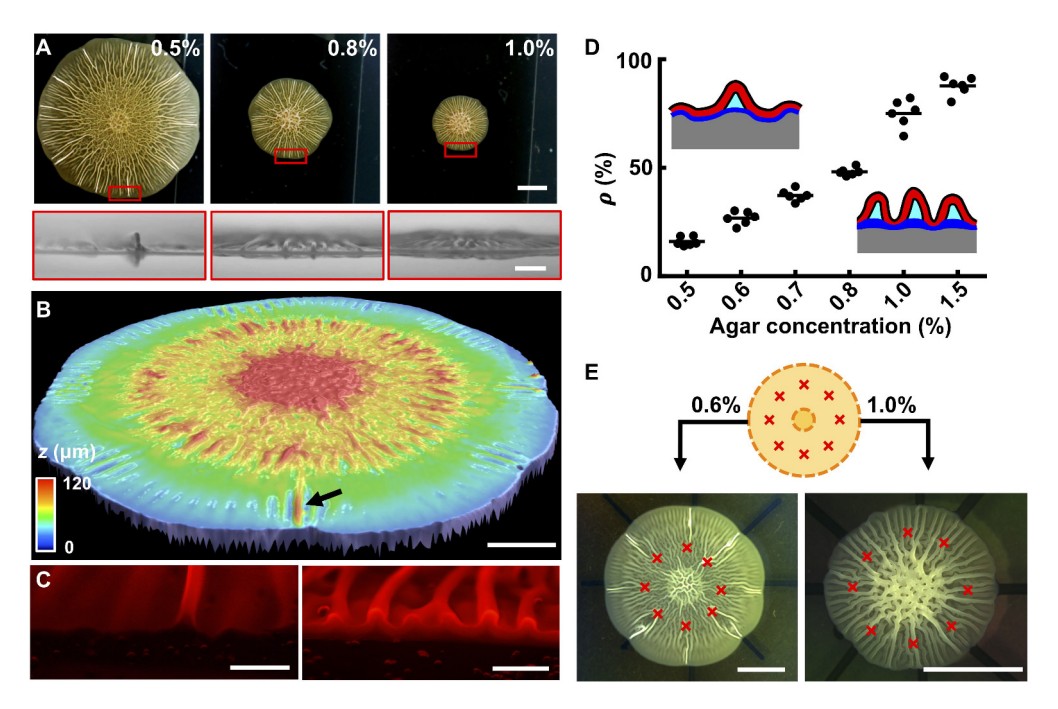

**Figure 3.** The biofilm wrinkling-to-delamination transition is controlled by adhesion energy. (**A**) Top (*top*) and side (*bottom*) views of biofilms on plates containing the designated concentrations of agar taken 10 hr after the onset of delamination. Scale bar: 5 mm (*top*) and 1 mm (*bottom*). (**B**) Surface topography of a biofilm grown on 0.5% agar at the onset of the wrinkling-to-delamination transition (36 hr). The arrow indicates a blister. Scale bar: 2 mm. (**C**) Cross-sectional views of rims of biofilms producing fluorescent mKate2, grown for 40 hr on plates containing 0.6% agar (*left*) and 1.0% agar (*right*). Scale bars: 0.5 mm. (**D**) Percentage ($\rho$) of blisters in all radially oriented features (wrinkles + blisters) versus agar substrate concentration for 2-day-old biofilms. The distinction between wrinkles and blisters is made on the basis of visual inspection. Insets: schematics showing how $\rho$ depends on substrate stiffness. Red with black outline, biofilms; gray, agar substrate; blue, residual layer; cyan, liquid between the blisters and the agar. (**E**) Biofilm growth on a substrate with defined defects. *Top*: schematic. Yellow denotes the growing biofilm. Red crosses denote the eight defects that were generated by manually making holes in the agar. *Bottom*: bright-field images of typical experiments using the setup shown in the schematic (*top*), for biofilms grown on plates with the designated agar concentrations. Scale bars: 5 mm.

DOI: https://doi.org/10.7554/eLife.43920.022

The following source data and figure supplements are available for figure 3:

**Source data 1.** Wrinkles and blisters in biofilms.
DOI: https://doi.org/10.7554/eLife.43920.023

**Figure supplement 1.** 3D topography of a biofilm blister before and after capillary peeling.
DOI: https://doi.org/10.7554/eLife.43920.024

**Figure supplement 1—source data 1.** Height profile of a large blister before and after capillary peeling.
DOI: https://doi.org/10.7554/eLife.43920.025

## Interfacial energy controls blister development dynamics and interactions between blisters

In conventional materials systems, a blister initially assumes a sinusoidal profile and then continues to grow in both width and height as the strain mismatch between the film and substrate increases (*Vella et al., 2009*). We wondered how blister width and height would develop in a living biofilm as the biofilm expands and accumulates strain mismatch. To examine this, we tracked isolated blisters by imaging the rim of the expanding biofilm (*Figure 2—figure supplement 1*). The width of each biofilm blister decreased while its height increased over time until the final width of the blister reached twice that of the thickness of the biofilm (*Figure 4A,B*). This final value for the blister width indicates that the two sides of the blister come into contact with one another. Subsequently, blisters

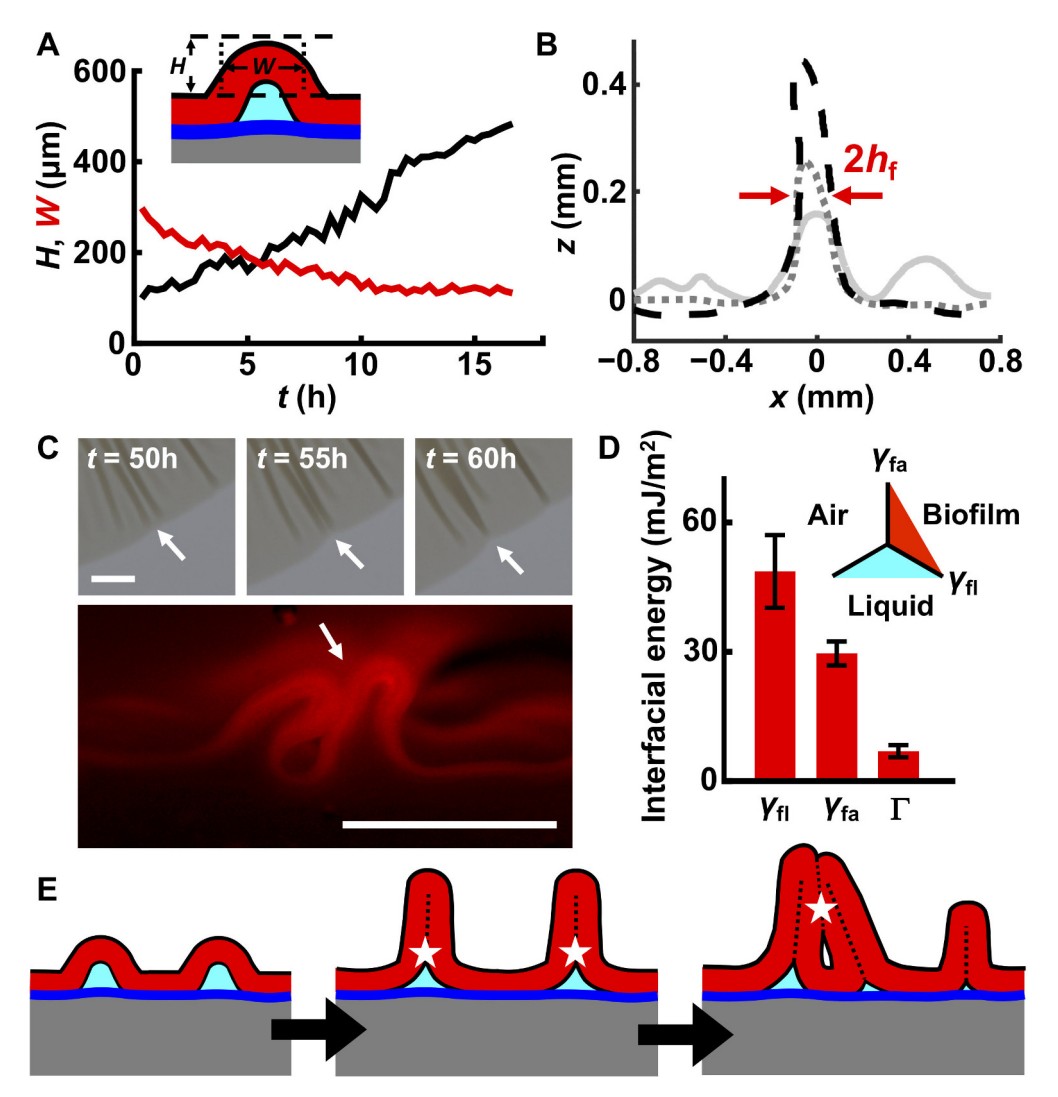

**Figure 4.** Interfacial energies control blister dynamics and interactions between blisters. (**A**) Time evolution of the height $H$ (black) and width $W$ (red) of a representative biofilm blister. Inset: schematic representation of a blister; color code as in **Figure 3D**. (**B**) Developing profile of a single blister, extracted from side view images at successive time points after delamination. Profiles are shown at 2.5 hr (gray line), 10 hr (gray dotted line) and 17.5 hr (black dashed line) after the onset of delamination. The distance between the red arrows corresponds to $W$, which, over time, approaches twice the biofilm thickness ($2h_f$). Regions near the blister become flatter as cell mass is pulled into the blister. Agar concentration: 0.4%. (**C**) Representative merging of adjacent blisters (white arrows) at specified times (*top*). Cross-section image from a biofilm producing fluorescent mKate2 reveals blister peak-to-peak contact (*bottom*; designated by the white arrow). Agar concentration: 0.7%. Scale bars: 1 mm (*top*) and 0.5 mm (*bottom*). (**D**) Interfacial energy of the biofilm–air interface $\gamma_{fa}$, biofilm–liquid interface $\gamma_{fl}$, and the adhesion energy between the biofilm and the substrate $\Gamma$ for WT *V. cholerae* biofilms. Data are represented as mean ± std with $n = 3$. *Inset*: schematic of different interfaces. (**E**) Schematic of blister development in a WT *V. cholerae* biofilm. White stars and dashed black lines denote interface annihilation events. For panels (**D**) and (**E**), the color code is the same as that in **Figure 3D**.

DOI: https://doi.org/10.7554/eLife.43920.026

The following source data and figure supplements are available for figure 4:

**Source data 1.** Blister formation and evolution dynamics and related interfacial energies in WT *V. cholerae* biofilms.
DOI: https://doi.org/10.7554/eLife.43920.027
**Figure supplement 1.** Characterization of pattern merging events.
DOI: https://doi.org/10.7554/eLife.43920.028

*Figure 4 continued on next page*

*Figure 4 continued*

**Figure supplement 1—source data 1.** Wavelength analysis over three days of biofilm development.
DOI: https://doi.org/10.7554/eLife.43920.029
**Figure supplement 2.** Analysis of the internal structures of biofilm blisters.
DOI: https://doi.org/10.7554/eLife.43920.030
**Figure supplement 3.** Bacterial cells residing in biofilm blisters are protected from antibiotics.
DOI: https://doi.org/10.7554/eLife.43920.031

continue to develop only in height. Moreover, large blisters suppress nearby wrinkles from delaminating (*Figure 4B*), presumably because the biofilm and the substrate can slide relative to one another such that a blister captures nearby compressed biofilm material, and in so doing, releases compressive stress in the vicinity. Neighboring blisters tend to merge during late stages of biofilm development (>48 hr), forming single dark features in the transmission images (*Figure 4C* (*top*) and *Figure 4—figure supplement 1*). Indeed, cross-sectional images reveal that head-to-head contact occurred (*Figure 4C* (*bottom*)).

The sequential biofilm blister dynamics described above involve the generation or annihilation of new or existing interfaces, which have energy penalties or payoffs. To understand the order of these events, we measured their interfacial energies in WT *V. cholerae* biofilms (*Yan et al., 2018*). They are: biofilm blister–liquid underneath, $\gamma_{fl} \sim 49$ mJ/m$^2$; biofilm blister–air above, $\gamma_{fa} \sim 30$ mJ/m$^2$; and the energy needed to separate the biofilm from the residual layer underneath, $\Gamma \sim 5$ mJ/m$^2$ (*Figure 4D*). This energy hierarchy determines the sequence through which interfaces are generated or annihilated (*Figure 4E*). First, compressive stress leads to delamination of the biofilm from the residual layer, forming a local blister. This step generates an additional high-energy interface between the blister and the liquid underneath it. To eliminate this high-energy interface, the two sides of the inner face of the blister come into contact with each other as the blister grows. Indeed, electron microscopy imaging of the cross-section of a blister shows this to be the case (*Figure 4—figure supplement 2*). After internal contact occurs, the blister can only develop in the vertical direction. However, blister growth enlarges the interface between the biofilm and the air. Subsequent merging of neighboring blisters (*Figure 4C*) eliminates biofilm–air interfaces, and in so doing, lowers the free energy of the entire system. An added benefit to the bacteria stems from these blister dynamics: cells in blisters are less susceptible to the lethal effects of antibiotics that diffuse in from the substrate than are cells residing in the base of the biofilm, presumably because cells in blisters are located further away from the antibiotic source (*Figure 4—figure supplement 3*).

If the above interpretations concerning the involvement of interfacial energy in blister development are correct, changing the relative magnitudes of the three interfacial energies should modulate blister dynamics, and, in turn, the global biofilm morphogenesis process. To test this idea, we deleted *bap1* and *rbmC*, which encode proteins that are responsible for cell-surface interactions and biofilm hydrophobicity (*Fong and Yildiz, 2007*; *Berk et al., 2012*; *Hollenbeck et al., 2014*). Rather than forming isolated blisters, when formed on soft agar substrates, the Δ*bap1*Δ*rbmC* biofilm exhibits a star-shaped morphology with flat regions between the facets of the stars (*Figure 5A* (*top*)) (*Yan et al., 2017*). The cross-section of a single facet shows that it consists of a group of congregated blisters (*Figure 5A* (*bottom*)). Curiously, in contrast to the WT blisters, in the mutant, only the external surfaces of neighboring blisters are in contact with one another, leaving the internal spaces under each blister intact. Indeed, transmission images show that multiple stripes exist within one facet, corresponding to multiple blisters (*Figure 5B*, *Figure 5—figure supplement 1*).

To rationalize the Δ*bap1*Δ*rbmC* blister dynamics, we measured the relevant interfacial energies (*Figure 5C*). The adhesion energy $\Gamma$ between the Δ*bap1*Δ*rbmC* biofilm and the substrate is below the detection limit, meaning that delamination occurs more easily in the Δ*bap1*Δ*rbmC* biofilm than in the WT biofilm. Indeed, Δ*bap1*Δ*rbmC* biofilm blisters emerge directly from the expanding flat film, skipping the wrinkling state (*Video 6*). Second, the relative order of interfacial energies changes in the mutant: $\gamma_{fl}$ approaches zero whereas $\gamma_{fa}$ is large, consistent with the hydrophilicity of the Δ*bap1*Δ*rbmC* biofilm (*Hollenbeck et al., 2014*). These alterations in interfacial energies have profound consequences for blister dynamics (*Figure 5D*). Instead of annihilating biofilm–liquid interfaces inside of the blisters, in the mutant, neighboring blisters prefer to collapse against each other, which eliminates the high-energy interface between the biofilm and the air. Indeed, during the

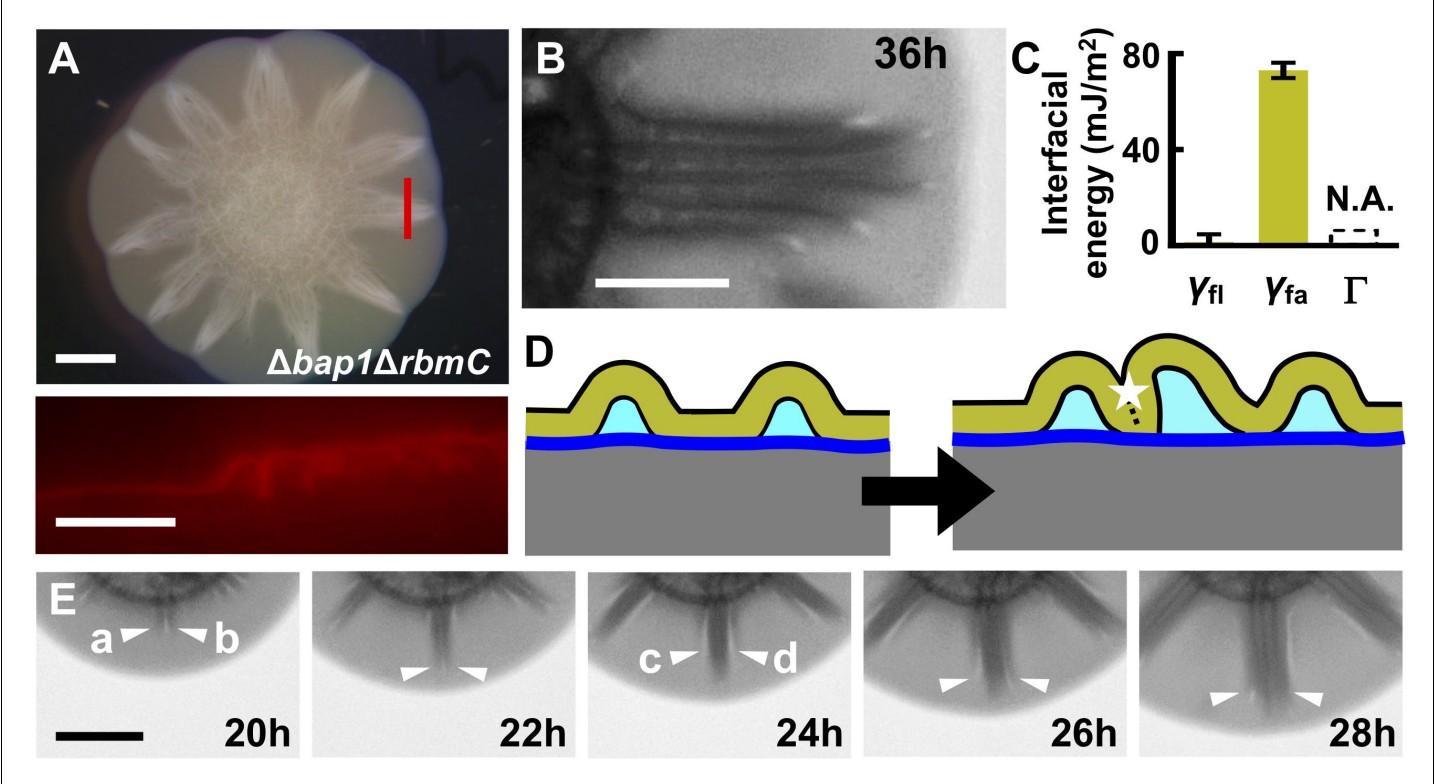

**Figure 5.** Morphogenesis of a mutant biofilm possessing altered interfacial energies. (**A**) Bright-field (*top*) and cross-sectional (*bottom*) images of a *V. cholerae* Δ*bap1*Δ*rbmC* mutant (abbreviated as Δ*BC* below) biofilm producing fluorescent mKate2, grown for 2 days on a 0.6% agar substrate. The red line in the *top* panel indicates the location of the cross-section used for the *bottom* panel. Scale bars: 2 mm (*top*) and 500 μm (*bottom*). (**B**) Close-up view of a star facet in a Δ*BC* biofilm grown on 0.6% agar for 36 hr. Scale bar: 1 mm. (**C**) Interfacial energies measured for the Δ*BC* biofilm. N.A. means too small to be measured. Data are represented as mean ± std with *n* = 3. (**D**) Schematic representations of Δ*BC* biofilm morphology development. Color code as in *Figure 3D*, except that yellow represents the Δ*BC* biofilm. (**E**) Transmission images of a section of a Δ*BC* biofilm growing on a 0.6% agar plate at the designated times. White arrowheads indicate emerging blisters. Four blisters (a–d) emerged during the time shown. Scale bar: 1 mm.
DOI: https://doi.org/10.7554/eLife.43920.032

The following source data and figure supplements are available for figure 5:

**Source data 1.** Interfacial energies of *V. cholerae* Δ*bap1*Δ*rbmC* mutant biofilms.
DOI: https://doi.org/10.7554/eLife.43920.033
**Figure supplement 1.** Interfacial energies determine the morphological features of the biofilm.
DOI: https://doi.org/10.7554/eLife.43920.034
**Figure supplement 1—source data 1.** Height profiles of WT *V. cholerae* and Δ*bap1*Δ*rbmC* mutant biofilms.
DOI: https://doi.org/10.7554/eLife.43920.035

development of the mutant biofilm, newly emergent blisters move towards, and ultimately join, existing blister groups (*Figure 5E*; *Video 6*). The triangular shape of each facet in the Δ*bap1*Δ*rbmC* biofilm is therefore a consequence of the merging of multiple blisters, whose ages and radial lengths decrease from the center to the edge of the aggregate.

## Mechanical instability and biofilm expansion feed back onto one another

We wondered whether the emergence of the 3D biofilm surface topography affected biofilm expansion in the growing plane. One common morphological feature of bacterial biofilms is their irregular petal-shaped 2D contours (*Videos 3* and *4*). We hypothesized that the evolution of contours could also be a consequence of blister formation. To quantify the contour undulation, we define the acircularity parameter $\alpha = P^2/4\pi A$, in which $P$ is the perimeter of the biofilm and $A$ is the area

(*Asally et al., 2012*). $\alpha = 1$ for a perfect circle. For a biofilm growing on soft agar (0.4%, *Figure 6A*), there is a sharp increase in $\alpha$ at $t_c$, the time at which the 3D surface morphology forms at the edge (*Figure 6—figure supplement*

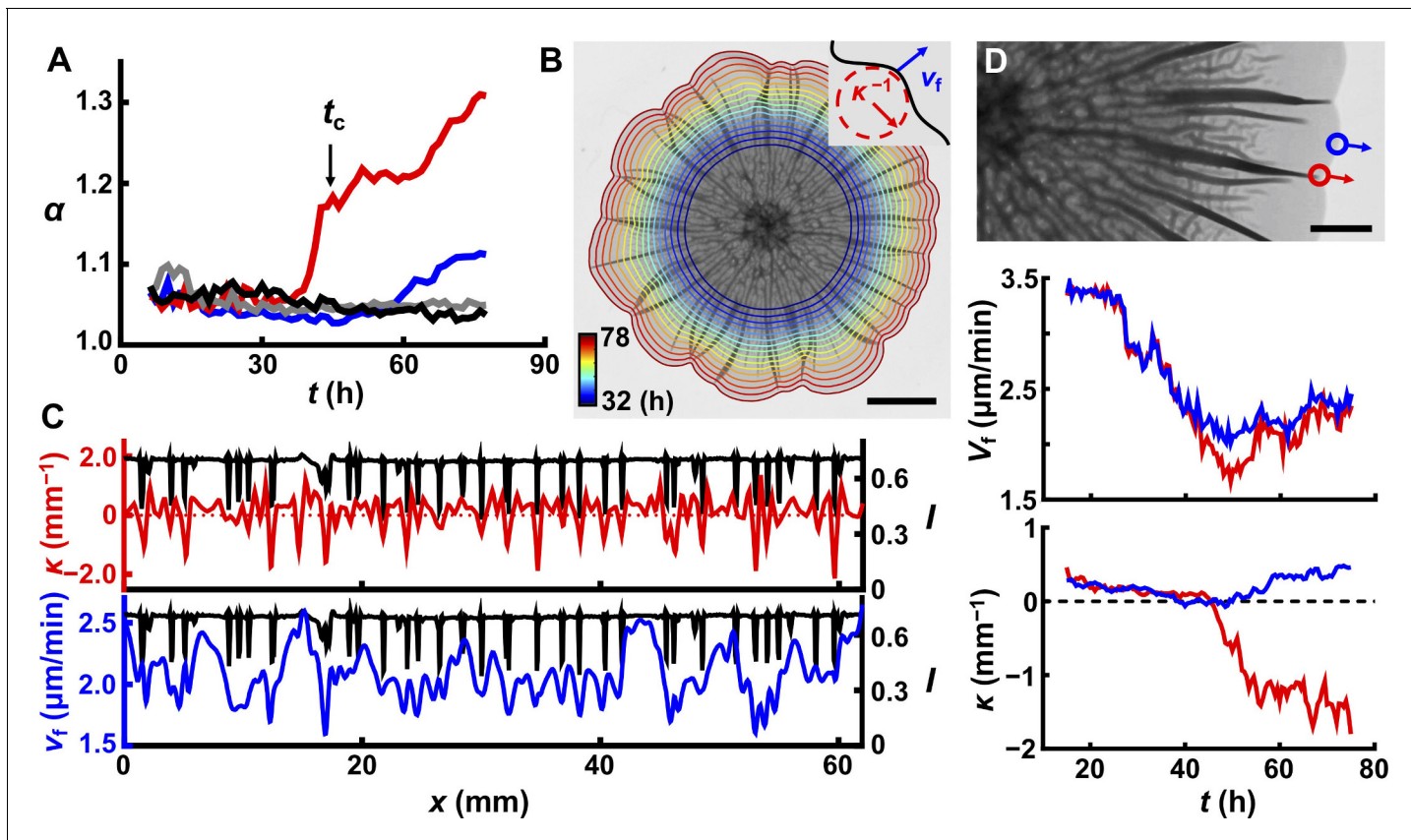

**Figure 6.** Delamination defines the overall biofilm contour. (**A**) Time evolution of acircularity index $\alpha$ (where $\alpha = P^2/4\pi A$, in which $P$ is the perimeter of the biofilm and $A$ is the area) of the biofilm contour. Two agar substrate concentrations are shown (0.4%, red; 1.0%, blue) for WT *V. cholerae* biofilms. The sharp upturn in $\alpha$ defines the critical time $t_c$. Biofilms lacking matrix ($\Delta vpsL$ mutant; 0.4%, gray) or possessing an unstructured matrix ($\Delta rbmA\Delta bap1\Delta rbmC$ mutant; 0.4%, black) remain circular. (**B**) Image of a WT *V. cholerae* biofilm grown on 0.7% agar 78 hr after inoculation, overlaid with the time evolution of the biofilm boundary. Colors correspond to the expanding boundary from 32 to 78 hr. Scale bar: 5 mm. *Inset*: schematic of local velocity $V_f$ and the inverse of local curvature $\kappa^{-1}$. (**C**) Transmitted light intensity profiles $I$ (black), $\kappa$ (red), and $V_f$ (blue) along the biofilm periphery from panel (**B**) at 60 hr. (**D**) *Top*: partial image of the biofilm shown in panel (**B**) at 75 hr. Red and blue dots denote two boundary points at the locations of a delaminated and a flat region, respectively. Arrows indicate boundary expansion. *Middle and bottom*: time evolution of $V_f$ and $\kappa$ of the designated time points during biofilm development. Scale bar: 2 mm.

DOI: https://doi.org/10.7554/eLife.43920.037

The following source data and figure supplements are available for figure 6:

**Source data 1.** Local curvature, velocity, and transmission image intensity, and acircularity for biofilm contour evolution dynamics.
DOI: https://doi.org/10.7554/eLife.43920.038

**Figure supplement 1.** Delamination triggers global and local slowdown of biofilm expansion and shapes the biofilm contour.
DOI: https://doi.org/10.7554/eLife.43920.039

**Figure supplement 1—source data 1.** Analysis of contour evolution and biofilm expansion dynamics.
DOI: https://doi.org/10.7554/eLife.43920.040

**Figure supplement 2.** Blister formation drives the overall biofilm contour.
DOI: https://doi.org/10.7554/eLife.43920.041

**Figure supplement 2—source data 2.** Local curvature, velocity, and transmission image intensity for biofilm contour evolution dynamics in a line geometry.
DOI: https://doi.org/10.7554/eLife.43920.042

*1*). To show that blisters are required for contour undulations, we tracked $\alpha$ for mutant biofilms lacking the matrix structural polysaccharide ($\Delta vpsL$) (*Figure 4—figure supplement 3*; *Hammer and Bassler, 2003*) or lacking matrix structural proteins ($\Delta rbmA\Delta bap1\Delta rbmC$) (*Figure 2—figure supplement 3C*; *Berk et al., 2012*; *Yan et al., 2017*). In both cases, the biofilm has no surface features and $\alpha$ remains close to 1 (*Figure 6A*).

To investigate the coupling between contour undulations and biofilm morphogenesis in the *z* direction, we followed the time evolution of growing biofilm borders in different geometries (*Figure 6B*, *Figure 6—figure supplement 2*). Visually, the indentations along the contours always correspond to the locations of large blisters. To quantify this finding, we measured the local curvature $\kappa$ and expansion velocity $V_f$ along the biofilm periphery (*Figure 6B,C*, *Figure 6—figure supplements 1* and *2*). Both $\kappa$ and $V_f$ are negatively correlated with the positions of blisters. Monitoring the evolution of a single blister and a nearby flat region shows a transient large difference in $V_f$ when the blister initially forms at the edge ($\sim$ 45 hr in this case; *Figure 6D*), which triggers the local contour indentation. The emergence of a blister creates an extra dimension into which newly produced biomass can be distributed, which causes local slowing in $V_f$, thus establishing the correlation between blister locations and negative local curvature. After this transient difference, $V_f$ becomes comparable for boundaries with and without blisters, and the local curvature reaches a steady value, provided that there is no nearby blister (*Figure 4—figure supplement 1*). In this steady state, the petal-like contour propagates radially without changing the overall shape of the contour. This explanation for the formation of the biofilm petal shapes suggests that contour undulations require non-homogeneous blister distribution along the biofilm rim and indeed, WT biofilms that are grown on stiff agar (>1.0%) remain nearly circular because they possess regularly and closely spaced blisters (*Figure 6A*, blue line). As additional evidence for the connection between blister formation and boundary undulation, we show that we can control the number and positions of the petals by specifying the positions of the blisters using patterned substrates (*Figure 3*, *Figure 6—figure supplement 2*). We conclude that the 3D surface topography that arises owing to mechanical instabilities caused by biofilm expansion feeds back to slow down expansion and drive contour evolution.

## Discussion

We show here that mechanical instabilities, including wrinkling and delamination, underlie biofilm morphogenesis. Moreover, differences in interfacial energies drive mechanomorphogenesis by dictating the creation or annihilation of new or existing interfaces. Finally, feedback between mechanomorphogenesis and biofilm expansion shapes the overall biofilm contour. Collectively, our findings concerning the connections between a biofilm's surface morphology and its mechanical and material properties suggest that new genes and/or new compounds that alter biofilm morphology by altering mechanics could be discovered and investigated to address biofilm-related problems.

Morphological patterns can certainly involve gene regulation programs. Nonetheless, we expect our mechanical instability findings in *V. cholerae* biofilms to apply to other systems — from bacteria to humans — because they reveal links between the specific material properties of the biological components and morphological transitions. Regarding bacterial systems, we have already commented on how localized cell death underpins pattern formation at the core of *Bacillus subtilis* biofilms (*Asally et al., 2012*). In fact, in light of our mechanomorphogenesis model, localized cell death can be viewed as a source of surface defects that functions to trigger delaminations, similar to the defined surface imperfections that drive delaminations shown in *Figure 3E*. Another example concerns biofilms of *Pseudomonas aeruginosa*, an opportunistic pathogen (*Costerton et al., 1999*). WT *P. aeruginosa* develops biofilms with a labyrinthine inner pattern surrounded by flat rims (*Madsen et al., 2015*). By contrast, *P. aeruginosa* mutants that are incapable of phenazine production ($\Delta phz$) form biofilm topography similar to those that we examined here for *V. cholerae* with disordered cores surrounded by radial features (*Dietrich et al., 2013*). We suggest that the mechanical principles uncovered here could also drive the morphological transitions in *P. aeruginosa* biofilms. The WT *P. aeruginosa* biofilm pattern occurs because cells at the biofilm center display upregulated matrix production (*Madsen et al., 2015*), whereas cells located at the periphery are downregulated for matrix production. In the case of the $\Delta phz$ mutant, all of the cells overproduce extracellular polysaccharides (*Madsen et al., 2015*), so we speculate that the $\Delta phz$ *P. aeruginosa* mutant forms peripheral radial wrinkles and subsequently delaminations because of the same

mechanical instability described here in *V. cholerae*. These examples illustrate how gene regulation and spatially differentiated cell physiology can be coupled to mechanical instability to promote biofilm surface morphologies.

Recent theoretical work on bacterial biofilms has considered mechanical instabilities. *Zhang et al. (2016)* used simulations to suggest that anisotropic growth coupled with wrinkling instability could generate the surface topography observed in bacterial biofilms, and most recently they considered the possibility of delamination (*Zhang et al., 2017*). *Wang and Zhao (2015)* introduced competition between adhesive and elastic energies and computed a phase diagram of the different modes of instability for a film–substrate system. These inspiring theories will be made more valuable by the inclusion of measured biophysical parameters and additional observations generated through experiments. For example, the thin intermediate residual layer that we discovered here is not accurately considered in biofilm simulations, but is required to explain the wrinkling instability in biofilms (*Figure 2D*). In addition, interfacial energies play a predominant role in driving the morphologies of biological materials that possess soft layers, whereas their roles are minor in classical mechanical systems (*Qi et al., 2011*). To date, contributions from interfacial energies have been suggested in contexts such as cell sorting in tissues (*Brodland, 2002*; *Foty and Steinberg, 2005*), but we are not aware of any work incorporating interfacial energies into mechanical instability models for morphogenesis. Future theoretical analyses can now incorporate measured parameters to understand the rich hierarchical dynamics and the history dependence of mechanomorphogenesis, taking into account biofilm viscoelasticity, interfacial energies, and the consequences of sliding and friction between the biofilm and the substrate (*Beroz et al., 2018*; *Peterson et al., 2015*).

Though more sophisticated, eukaryotic organisms often employ similar mechanical instability principles to generate fascinating morphologies. Thus, our findings for biofilms are potentially generalizable and relevant for studies of development in higher organisms (*Kim et al., 2015*). A close analogy is presented by cerebellum development. The cerebellum possesses a thin, soft layer of Purkinje cells that is sandwiched between the rapidly growing external granular layer and the slow-growing internal granular layer (*Lejeune et al., 2016b*). Through wrinkling instabilities, the cerebellum develops finely spaced parallel grooves called folia. This hard-soft-hard geometry and the associated wrinkling instabilities directly mirror the configuration that we discovered in *V. cholerae* biofilms. Hence, our work suggests that exploiting mechanical principles to drive key morphogenic events is ancient: it occurs in bacteria, and evolution, as is often the case, has reused prokaryotic processes and principles in eukaryotes. In summary, biofilms represent an intriguing and highly tractable model system to investigate the general role of mechanical forces in morphogenesis, and they provide a convenient system for morpho-engineering.

# Materials and methods

**Key resources table**

| Reagent type (species) or resource | Designation | Source or reference | Identifiers | Additional information |
|---|---|---|---|---|
| Strain, strain background (*E. coli*) | S17 λ-*pir* | *de Lorenzo and Timmis, 1994* | | Wild type |
| Strain, strain background (*V. cholerae*) | C6706*str2* | *Thelin and Taylor, 1996* | | El Tor wild type |
| Strain, strain background (*V. cholerae*) | JY283 | *Yan et al., 2017* | | $vpvC^{W240R}$ $\Delta pomA$ (denoted WT) |
| Strain, strain background (*V. cholerae*) | JY285 | *Yan et al., 2017* | | $vpvC^{W240R}$ $\Delta pomA\Delta bap1\Delta rbmC$ |
| Strain, strain background (*V. cholerae*) | JY286 | *Yan et al., 2017* | | $vpvC^{W240R}$ $\Delta pomA\Delta rbmA\Delta bap1\Delta rbmC$ |

*Continued on next page*

*Continued*

| Reagent type (species) or resource | Designation | Source or reference | Identifiers | Additional information |
|---|---|---|---|---|
| Strain, strain background (*V. cholerae*) | JY287 | *Yan et al., 2017* | | $vpvC^{W240R}$ $\Delta pomA\Delta vpsL$ |
| Strain, strain background (*V. cholerae*) | JY370 | *Yan et al., 2017* | | $vpvC^{W240R}\Delta pomA$ $lacZ$:$P_{tac}$-*mKate2*:*lacZ* |
| Strain, strain background (*V. cholerae*) | JY376 | *Yan et al., 2017* | | $vpvC^{W240R}\Delta pomA$ $\Delta vpsL$ $lacZ$:$P_{tac}$-*mKate2*:*lacZ* |
| Strain, strain background (*V. cholerae*) | JY395 | This study | | $vpvC^{W240R}\Delta pomA$ $\Delta bap1\Delta rbmC$ $lacZ$:$P_{tac}$-*mKate2*:*lacZ* |
| Recombinant DNA reagent | Plasmid: pKAS32 | *Skorupski and Taylor, 1996* | | Suicide vector, $Amp^R$ $Sm^S$ |
| Recombinant DNA reagent | Plasmid: pNUT144 | *Drescher et al., 2014* | | Suicide vector, $Amp^R$ $Kan^R$ $Sm^S$ |
| Recombinant DNA reagent | Plasmid: pNUT157 | *Drescher et al., 2014* | | pNUT144 $vpvC^{W240R}$ |
| Recombinant DNA reagent | Plasmid: pCMW112 | *Hammer and Bassler, 2003* | | pKAS32 $\Delta vpsL$ |
| Recombinant DNA reagent | Plasmid: pCN004 | *Nadell and Bassler, 2011* | | pKAS32 $lacZ$:$P_{tac}$-*mKate2*:*lacZ* |
| Recombinant DNA reagent | Plasmid: pCN007 | *Nadell et al., 2015* | | pKAS32 $\Delta rbmA$ |
| Recombinant DNA reagent | Plasmid: pCN008 | *Nadell et al., 2015* | | pKAS32 $\Delta rbmC$ |
| Recombinant DNA reagent | Plasmid: pCN009 | *Yan et al., 2016* | | pKAS32 $\Delta bap1$ |
| Recombinant DNA reagent | Plasmid: pCDN010 | *Nadell et al., 2015* | | pKAS32 $\Delta pomA$ |
| Software, algorithm | MATLAB and Image Processing Toolkit | Mathworks, 2015 | | https://www.mathworks.com/products/matlab.html |
| Software, algorithm | PRISM version 6.07 | GraphPad, 2015 | | https://www.graphpad.com/scientific-software/prism/ |
| Software, algorithm | Image composite editor version 2.0.3 | Microsoft, 2015 | | https://www.microsoft.com/en-us/research/project/image-composite-editor/ |
| Software, algorithm | Gmsh version 3.0.6 | *Geuzaine and Remacle, 2009* | | https://gmsh.info |
| Software, algorithm | Paraview version 5.5.0 | *Ahrens et al., 2005* | | https://www.paraview.org/ |
| Software, algorithm | FEniCS version 2017.2.0 | *Alnæs et al., 2015* | | https://fenicsproject.org/ |
| Software, algorithm | DigiCamControl software version 2.0.72.0 | DigiCamControl, 2015 | | http://digicamcontrol.com/ |
| Software, algorithm | Leica Map Start version 7.4.8051 | Leica, 2017 | | https://www.leica-microsystems.com/products/microscope-software/details/product/leica-map/ |

*Continued*

| Reagent type (species) or resource | Designation | Source or reference | Identifiers | Additional information |
|---|---|---|---|---|
| Software, algorithm | ImageJ and freehand line selection tool | NIH | | https://imagej.nih.gov/ij/ |
| Software, algorithm | RheoPlus version 3.40 | Anton Paar, 2008 | | |
| Other | LB broth, Miller | ThermoFisher | Cat# BP1426-2 | |
| Other | Bacto agar | VWR | Cat# 214030 | |
| Other | O.C.T. agent | Tissue-Tek, Sakura | Cat# 4583 | |
| Other | Silicone oil, 5 cSt | Sigma Aldrich | Cat# 317667 | |
| Other | Glass beads, acid washed, 425 – 600 µm diameter | Sigma Aldrich | Cat# G8772 | |
| Other | MP Biomedicals Roll & Grow Plating Beads, 4 mm in diameter | ThermoFisher | Cat# MP115000550 | |
| Other | BD PrecisionGlide needles (0.6 mm × 2.5 mm) | Sigma Aldrich | Cat# Z118044 | |
| Other | EMD Millipore, 25 mm in diameter | Sigma Aldrich | Cat# VSWP02500 | |
| Other | SytoX Green Nucleic Acid Stain | ThermoFisher | Cat# S7020 | |
| Other | Wheat Germ Agglutinin Sampler Kit | ThermoFisher | Cat# W7024 | |
| Other | Higgins Black India Ink | | | |
| Other | Physica MCR 301 shear rheometer | Anton Paar, 2008 | | |
| Other | Nikon D3300 SLR digital camera with DX Zoom-Nikkor 18-55 mm lens | Amazon | | https://www.amazon.com/Nikon-1532-18-55mm-3-5-5-6G-Focus-S/dp/B00HQ4W1QE/ref=sr_1_3?ie=UTF8&qid=1492108083&sr=8-3&keywords=D3300&th=1 |
| Other | Huion L4S light box | Amazon | | https://www.amazon.com/Huion-L4S-Light-Box-Illumination/dp/B00J0UUHPO |
| Other | Sigma 105 mm macro lens for Nikon DSLR camera | Amazon | | https://www.amazon.com/Sigma-258306-105mm-Macro-Camera/dp/B0058NYW3K/ref=sr_1_sc_3?ie=UTF8&qid=1485483491&sr=8-3-spell&keywords=sigma+macroles |
| Other | Leica stereoscope model M205 FA | Leica | | |

*Continued on next page*

*Continued*

| Reagent type (species) or resource | Designation | Source or reference | Identifiers | Additional information |
|---|---|---|---|---|
| Other | Leica DCM 3D micro-optical system | Leica | | https://www.leica-microsystems.com/products/light-microscopes/upright-microscopes/details/product/leica-dcm-3d/ |
| Other | VR3200 wide-area 3D measurement system | Keyence | | https://www.keyence.com/products/measure-sys/3d-measure/vr-3000/models/vr-3200/index.jsp |
| Other | FEI XL 30 FEG-SEM | FEI | | https://iac.princeton.edu/equipment.html |
| Other | Millrock Technology, BT85A-A | Millrock | | https://www.millrocktech.com/ |
| Other | VCR IBS/TM200S ion beam sputterer | VCR | | https://iac.princeton.edu/equipment.html |

## Bacterial strains

All of the *V. cholerae* strains used in this study are derivatives of *V. cholerae* O1 biovar El Tor strain C6706str2 (*Thelin and Taylor, 1996*), harboring a missense mutation in the *vpvC* gene (VpvC W240R) (*Beyhan and Yildiz, 2007*). Bacterial cultures were grown at 37°C under constant shaking in standard lysogeny broth (LB) medium. Genetic engineering of *V. cholerae* was performed using allelic exchange with pKAS32 (*Skorupski and Taylor, 1996*). All plasmids used in the current study have been reported previously (see 'Key resources table'). pKAS32-derived plasmids were introduced into *V. cholerae* by conjugation with *Escherichia coli* S17 $\lambda$-*pir* (*de Lorenzo and Timmis, 1994*), selection on plates containing ampicillin (100 mg/L) and polymyxin B (6 mg/L), and subsequent counterselection on plates containing streptomycin (500 mg/L). Deletions were verified by PCR and phenotypic analysis. The constitutive *mKate2* gene (*Shcherbo et al., 2007*) is driven by $P_{tac}$ and was inserted into the *V. cholerae* chromosome at the *lacZ* locus (as previously described) with X-Gal (50 mg/L) present in the counterselection step (*Nadell and Bassler, 2011*).

## Biofilm growth

### Biofilm growth on agar plates

LB medium solidified with different percentages of agar was used as the solid support to grow biofilms. *V. cholerae* strains were streaked onto LB plates containing 1.5% agar and grown at 37°C overnight. Individual biofilms were selected and inoculated into 3 mL of LB liquid medium containing ~ 10 glass beads (MP Biomedicals Roll and Grow Plating Beads, 4 mm diameter) and the cultures were grown with shaking at 37°C to mid-exponential phase (5–6 hr). Subsequently, the cultures were mixed by vortex to break clusters into individual cells, $OD_{600}$ was measured, and the cultures were back-diluted to an $OD_{600}$ of 0.5. 1 μL of these preparations were spotted onto prewarmed agar plates. Subsequently, the plates were incubated at 37°C. Typically, four biofilms were grown per agar plate. For time-lapse imaging, one or two biofilms were grown on each plate.

### Biofilm growth on substrates with defined defects

On prewarmed agar plates, syringe needles (BD PrecisionGlide needles, 0.6 mm × 2.5 mm) were used to punch holes at eight locations, equally separated by 45° around a circle. Marks were made on the bottoms of the Petri dishes to guide our eyes for placement of holes in the agar surface. 1 μL of *V. cholerae* cultures at $OD_{600}$ = 0.5, prepared as described in the preceding paragraph, were spotted at the center of the circle. The diameter of the circle was ~ 1 cm for biofilms grown on 0.6% agar and ~ 0.6 cm for biofilms grown on 1.0% agar. Different circle diameters were used to accommodate the differently sized biofilms that form on soft and stiff agar, and to guarantee that, in both cases, when biofilms expanded to cover the pre-defined defects, they remained flat. Following

biofilm growth, the positions of these defects were inferred from the marks drawn on the bottoms of the Petri dishes.

## Biofilm growth in a line geometry

A *V. cholerae* culture at $OD_{600} = 0.5$ was prepared as described above. A sterile razor blade was carefully dipped into this culture and dried in air for 1 min. The razor blade was gently touched to the surface of a prewarmed agar plate to initiate biofilm growth.

## Biofilm growth at the liquid–air interface

First, a biofilm was grown for 24 hr following the procedure describe above. Subsequently, 25 mL of LB medium was gently added from the edge of the agar plate. When the liquid reached the biofilm, the liquid lifted the biofilm off the substrate by capillary force.

## Biofilm imaging

### Bright-field imaging

Biofilms were imaged with a Leica stereoscope in the reflective (bright field) mode. For biofilms larger than the field of view, multiple overlapping images were acquired manually (3 by 3 or 3 by 2) at different locations in the biofilm. Images from multiple locations in biofilms were stitched together with the Image Composite Editor software from Microsoft to yield the full images of the biofilms while preserving the original resolution. Raw images from the stereoscope contain iridescence as the result of reflections from agar, which were removed by setting the color saturation to zero (i.e. converting to black-and-white images).

### Transmission imaging

A custom transmission imaging setup was built in a 37°C environmental room to follow biofilm growth. Briefly, an agar plate containing the inoculum was placed on an LED illumination pad (Huion L4S Light Box) and imaged with a Nikon D3300 SLR camera equipped with a Sigma 105 mm F2.8 Macro Lens. The entire setup was covered to exclude light. The camera was controlled using Digi-CamControl software. Imaging was started 5 hr after inoculation, at which time the camera was capable of focusing on the growing biofilm. Imaging was performed automatically every 15 min for 3 d. The growth of the biofilm floating at the air–liquid interface was monitored with images acquired at 5 min intervals.

### Side view imaging

A similar setup to the one described in the preceding paragraph was used to image biofilms from the side, with the following changes. First, the LED illumination pad was placed on the side so that the camera received scattered light from the biofilm surface. Second, an additional camera (Nikon D3300 SLR equipped with DX Zoom-Nikkor 18–55 mm lens) was also placed on the side of the biofilm, at an ~ 90° angle with respect to the first optical path. To remove the optical obstruction from the wall of the agar plate, an imaging window (~ 1 cm × 1 cm) was created using a hot razor blade. Imaging started immediately before the onset of the wrinkling-to-delamination transition, and the time interval between images was 5 min. From time to time, the focus in the side view was adjusted manually.

### 3D optical profiling

Biofilms were imaged with a Keyence VR-3200 optical profiler using a telecentric multi-triangulation algorithm. Subsequent analyses related to obtaining the 3D profiles of biofilms were performed with the Keyence Analyzer software. In brief, noise was first removed from the raw data using the built-in function in the Keyence Analyzer software to give smooth, continuous surface profiles. Surfaces corresponding to agar were excluded by setting upper and lower height thresholds. 3D views of biofilms were rendered with a built-in function in the software. The corresponding line profiles were extracted along an arc centered at the center of the biofilm.

## Cross-sectioning of biofilms

Biofilms of *V. cholerae* strains expressing *mKate2* were grown on agar plates as described above. Where indicated, 0.5 μM SytoX Green Nucleic Acid Stain (ThermoFisher) was added to the agar to stain dead cells. The region of the agar substrate containing a biofilm (~ 2.5 cm × 2.5 cm) was removed and transferred to an empty petri dish. O.C.T. agent (Tissue-Tek, Sakura) was applied to the surface of the biofilms, and the entire Petri dish was rapidly dipped into a dry ice–ethanol mixture to solidify the O.C.T. agent together with the biofilm. Razor blades were used to cut through the solidified samples. Samples with exposed cross-sections were immediately transferred to a homemade T-shaped sample holder and kept frozen in a dry ice–ethanol mixture. These samples were transferred to a Leica stereoscope and imaged in bright-field mode or in fluorescent mode with an mCherry or GFP filter set.

## Rheological measurements

### Shear rheology of biofilms

All rheological measurements were performed with a stress-controlled shear rheometer (Anton Paar Physica MCR 301) at 37°C. For each measurement, 100–960 biofilms were collected with a pipette tip or a razor blade and transferred onto the lower plate of the rheometer. After sandwiching the biofilm cells between the upper and lower plates with a gap size of 0.5 mm, silicone oil (5 cSt at 25°C, Sigma Aldrich) was applied to surround the biofilm. Sandblasted surfaces were used for both the upper and lower plates to avoid slippage at the boundary. Oscillatory shear tests were performed with increasing amplitudes of the oscillatory strain $\varepsilon'$ from 0.01 to 2000% at a fixed frequency of 6.28 rad/s. The storage modulus $G'$ was extracted with the RheoPlus software as a function of $\varepsilon'$. To extract the plateau shear moduli of biofilms, segmented linear fittings were applied to $G'$-$\varepsilon'$ curves on a log-log scale. $G'$ varies minimally in the plateau region. We used the fitted $G'$ value at $\varepsilon' = 1\%$ as the modulus of the biofilm $G_f$. All rheological properties of the biofilm remained roughly constant for at least 48 hr.

### Shear rheology of agar

LB medium containing different agar concentrations was freshly prepared in 100 mL bottles. The semi-solid medium was heated in a microwave, cooled to ~ 55°C, and added (2 mL) to the lower plate of the rheometer preheated to 60°C. The heated agar solution was subsequently sandwiched between the two rheometer plates with a gap size of 0.5 mm and sealed with silicone oil. The preparation was cooled to 22°C using a cooling rate of 1°C/min. Subsequently, the solid agar was heated to 37°C for measurement. This procedure mimics the sequence of events that agar plates were exposed to during preparation and biofilm growth. Smooth surfaces with TrueGap technology were used. Oscillatory shear tests were performed in the linear elastic region at a fixed frequency of 6.28 rad/s. For data obtained with agar, we averaged 10–20 points in the plateau region of the $G'(\varepsilon')$ curve to give $G_s$.

### Poisson ratio measurement

The Poisson ratio $\nu$ of the biofilm was estimated by compressing the biofilm in the vertical direction and measuring its bulk modulus. Briefly, a home-built hollow cylinder made of polytetrafluoroethylene with a diameter of 25.5 mm was placed between two parallel plates of a rheometer. The biofilm was loaded into the cylinder to fill its volume. The upper plate of the rheometer (with a diameter of 25 mm) was subsequently lowered with a constant velocity (of between 8 mm/s and 12 mm/s). During this measurement, the shaft does not rotate, but rather acts as a piston to measure the normal force. Using the relationship between normal force and shaft displacement, we calculated the bulk modulus $K$ of the biofilm to be ~ 130 kPa; much larger than the shear modulus $G'$. From these data, we could calculate the Poisson's ratio $\nu = (3K - 2G') / 2(3K + G') \approx 0.495$, close to the incompressible limit ($\nu = 0.5$).

## Biofilm thickness measurements

The surface profiles of biofilms grown for 48 hr were analyzed with a Leica DCM 3D Micro-optical System. A 10× objective was used to image a 3 mm x 3 mm region covering roughly one quarter of the biofilm, with a $z$ step size of 2 μm. To measure the thickness of the residual layer, agar plates

containing biofilms were slowly vertically lowered into water to peel the biofilms from the substrate. The entire agar plate was allowed to air dry for 5–10 min to remove liquid remaining from the peeling step. After drying, the above analysis procedure was performed to measure the thickness of the residual layer.

The total thickness of the biofilm $h$ and the thickness of the residual layer $h_r$ were measured using Leica Map software. A three-point flattening procedure was first performed on the agar surface to level the image. Next, line profiles were generated at three different locations spanning the agar surface to the surface of the biofilm or the residual layer. An automatic step-size detection procedure was performed with a built-in function in the software to extract $h$ or $h_r$. The three measured values were averaged to give the value for one biological replicate. The biofilm thickness $h_f$ was obtained by $h_f = h - h_r$.

## SEM sample preparation and imaging

Biofilms were grown on 0.6% agar plates for 2 days as described above. The region of the agar containing a biofilm ($\sim$ 2 cm $\times$ 2 cm) was separated from the remainder of the agar plate, transferred to a piece of glass, and placed horizontally in a 50 mL conical tube and frozen at $-80°C$ overnight followed by overnight lyophilization (Millrock Technology, BT85A-A). The biofilm samples were sliced with a razor blade to expose blisters, sputter-coated with a 5 nm layer of Pd (VCR IBS/TM200S ion beam sputterer), adhered to an upright SEM stub with conductive tape, and imaged with a scanning electron microscope (FEI XL30 FEG-SEM).

## Characterization of biofilm residual layers

### Measurement of colony-forming units

Biofilms grown for 2 days were peeled off of agar substrates using a phosphate-buffered saline solution (PBS) as described previously (*Yan et al., 2018*). The floating biofilms were collected with clean pipette tips and the corresponding residual layers were removed from the agar using a sterile razor blade. All samples were transferred to 1.5 mL microcentrifuge tubes containing 1 mL PBS and $\sim$ 0.2 mL small glass beads (acid-washed, 425–600 μm diameter, Sigma), vigorously mixed by vortex for 15 min at 37°C to break apart aggregates, serially diluted in PBS, and plated onto LB plates. The LB plates were incubated overnight at 37°C and subsequently assessed for colony forming units (CFU). Four biological replicates were measured, each with two technical replicates. Raw CFU values were normalized by the volume of each biofilm and residual layer, calculated from the radius and thickness of each biofilm and residual layer, respectively.

### India Ink staining

Biofilms grown for 2 days were peeled off of agar substrates with PBS as described above. 1 mL of Higgins Black India ink solution (10% in PBS) was added to the agar to cover the area containing an intact biofilm or a residual layer, and the preparation was air-dried at room temperature for 30 min. The stained residual layer was subsequently imaged with a Leica stereoscope in the bright-field mode.

## Antibiotic killing assay

Biofilms of *V. cholerae* strains constitutively expressing *mKate2* were inoculated onto semipermeable membranes (EMD Millipore VSWP02500) that had been placed on top of 0.6% agar. The plates were incubated at 37°C for 2 days. The semipermeable membranes were gently removed from the agar surface using tweezers, and subsequently floated at room temperature overnight on top of 3 mL LB medium containing 1.7 μM SytoX Green stain with or without 50 μg/mL tetracycline.

## Biofilm image analyses

Image analyses were performed with custom codes written in MATLAB and with ImageJ software. Raw transmitted light image data were first converted into intensity images. From the pixel intensity distributions, we identified the peak with the highest intensity $I_b$ and used it as background. We set the minimum intensity $I_{min} = 0$ and the average background intensity $I_b = 0.9$ to standardize the contrast of the images. Images were then smoothed with a median filter. From the intensity distribution, we also identified the intensity value $I_V$ of the valley immediately adjacent to the background peak

and used it as the thresholding value to binarize the image (using a built-in thresholding function in MATLAB). We separated the biofilm object $F$ from the background. We used the image of each biofilm at $t$ = 12 hr after inoculation to define the center $O_F$ for all time points. When mutations affecting biofilm morphology arose, they were manually excluded from the image analysis.

To quantify variations in the amplitudes of biofilm morphological features, we extracted the intensity profiles $I^E(\theta)$ along a circle near the biofilm edge. We use a built-in function in MATLAB to identify the positions and the prominence $\Delta I_p$ of the peaks in $-I^E(\theta)$. We set the minimum peak prominence to be 0.02 to eliminate noise.

To extract the periodicity of the wrinkling or delamination pattern, we tracked the time evolution of these patterns from images. For wavelength analysis, we applied fast Fourier transformation (FFT) to intensity functions $I^r(\theta)$ in a ring at time $t$ and radial coordinate $r$, and identified $N(r,t)$ from the peak frequency in the power spectrum. We also verified the values by autocorrelation and manual counting. We plotted all data from different time points and fitted them with a linear function $N(r)$ $=2\pi r/\lambda$ to obtain the intrinsic wavelength $\lambda$. The radial coordinate at which $N$ decreases to zero was defined as $R_p$. For images of biofilms grown in a line geometry, several values of $N$ were extracted from multiple lines at different distances from the central line, averaged, and subsequently used to extract $\lambda$.

For contour analyses, we first obtained the biofilm object $F$ from the binarized image. From the binarized object $F$, we extracted the perimeter $P$ and the area $A$ of region $F$. At each time point, we calculated the acircularity $\alpha$ as $\alpha = P^2/4\pi A$. To define the radii for biofilms that were not strictly circular, we used $<R_f> = <|r_i-r_O|>_i$, averaged over all the points $r_i$ on the circumference $\partial F$. $<R_f>$ was then calculated over time to give $<R_f(t)>$ versus $t$. Segmented linear regression with two segments was used to quantify the expansion velocity of the biofilm $<V_f>$ before and after the critical time $t_c$ and to define the critical time itself.

To capture local curvature $\kappa$ and expansion velocity $V_f$, the smoothed boundary $\partial F$ was locally approximated by quadratic polynomials $r_{i,2}(t)$ at $r_i$. The parametrized curve $x_{i,2}(t)$ and $y_{i,2}(t)$ allowed us to calculate the analytical curvature $\kappa_i$ and normal $n_i$ locally using the weighted central difference. Coarse-grained contours at time points $t$ and $t+\Delta t$ were then connected by joining $r_i(t)$ to its nearest neighbor $r_i(t+\Delta t)$ in $\partial F_{t+\Delta t}$, yielding local velocities $V_{f,i} = |r_i(t+\Delta t) - r_i(t)|/\Delta t$.

To analyze the side-views of blisters, blister contours were manually extracted with ImageJ software and then smoothed. The baseline of the blister was obtained by averaging the $z$ coordinate of the left and right bottom region of the blister. The blister height $H$ was calculated as the distance between the peak of the blister to the baseline. The width of the blister $W$ was measured at half of the blister height.

## Theoretical modeling procedures

We adapted a trilayer model from previous work (*Lejeune et al., 2016b*), and modeled the biofilm system with the following three elastic components: the biofilm (top), the residual layer (middle), and the agar substrate (bottom) denoted with subscripts f, r, and s, respectively. *V. cholerae* biofilms harbor an active growing top cell layer and a dead cell layer underneath (*Figure 4—figure supplement 2*). The live and dead cell layers are connected to each other, and they were removed together for our mechanical measurements; so in the model, we do not distinguish between the two and we treat them as a single biofilm layer. Biofilm and residual layers were modeled as thin elastic sheets with thickness $h_f$ and $h_r$, whereas the agar substrate was modeled as an elastic body with a thickness $h_s$, much larger than that of the other two layers. The relevant scale for the continuum model is about the thickness of the film (>50 μm). Therefore, we could neglect potential structural and materials heterogeneities in the biofilm, which exist on a much smaller scale (~ 5 μm, see *Yan et al., 2018*). The shear modulus and Poisson's ratio of the materials are denoted by $G$ and $\nu$, respectively. For theoretical calculations, we treated all three layers as incompressible materials and hence, $\nu = 0.5$ (see the above experimental measurements). In the simulation, the residual layer grows at the same rate as the biofilm layer, while the substrate does not grow (as confirmed by comparing the locations of the edge of a biofilm and the residual layer; see *Figure 3—figure supplement 1*). This growth difference induces a strain mismatch $\varepsilon$ between the biofilm/residual layer and the substrate.

Following previous studies (*Lejeune et al., 2016a*), we applied the Föppl-von Kármán equation to the biofilm model. Assuming a sinusoidal profile of the surface undulations, we can write the longitudinal stress $S$ in the film as:

$$S(n) = \frac{G_\mathrm{f} h_\mathrm{f}^2 n^2}{3} + \frac{\tilde{K}}{h_\mathrm{f} n^2} ,$$

where $n$ is the wave number and $\tilde{K}$ is the combined stiffness of the residual layer and the substrate layer:

$$\tilde{K} = \frac{4 G_\mathrm{s} n}{n h_\mathrm{r}(G_\mathrm{s}/G_\mathrm{r} - 1) + 2} ,$$

and the effective substrate modulus of the composite substrate can be calculated by $G'_s = \tilde{K} h'_s$ in which $h'_s$ is the total depth of the strained region (see *Lejeune et al., 2016a* for details). By numerically solving the nonlinear equation d$S$/d$n$ = 0, we determined the minimal critical value of $S$ for mechanical instability and the corresponding $n$ gives the critical wavenumber $n_\mathrm{cr}$. The wavelength at the onset of wrinkling was then calculated as $\lambda_\mathrm{cr} = 2\pi/n_\mathrm{cr}$. The critical stress and strain were obtained by $S_\mathrm{cr} = S(n_\mathrm{cr})$ and $\varepsilon_\mathrm{cr} = S_\mathrm{cr}/3G_\mathrm{f}$, respectively. Theoretical predictions from the bilayer model can simply be calculated by setting $G_\mathrm{s} = G_\mathrm{r}$.

The model described above, despite assuming only small strains, accurately predicted the wavelength and critical stress/strain for finite strains (*Lejeune et al., 2016a*). We verified that the analytical predictions were in reasonable agreement with results obtained from finite element simulations.

The only unknown parameter in the model is the shear modulus of the residual layer $G_\mathrm{r}$, which is difficult to probe experimentally. Therefore, we treated $G_\mathrm{r}$ as the only fitting parameter. We used $h_\mathrm{r}/h_\mathrm{f}$ = 0.3 as an average value from the relevant experimental data and fit the model against the experimental data for wavelength versus stiffness contrast between the biofilm and the agar substrate. Fitting was carried out by minimizing the least-square error between the theoretically predicted and the experimentally measured wavelengths. A bisection method was employed that converged in fewer than 10 iterations.

## Computational modeling procedures

A plane-strain computational model was developed to take into account growth, large deformations, and the nonlinear elasticity of the system. We considered the same planar three-layer structure as above. According to finite strain theory, we define the deformation gradient tensor as $F_{ij} = \partial x_i / \partial X_j$, where $x_i$ and $X_i$ denote the coordinates in the deformed and undeformed configurations, respectively (*Ogden, 1997*). To incorporate the effect of growth, we further introduced the decomposition of the deformation tensor $\mathbf{F} = \mathbf{F}_\mathrm{e}\mathbf{F}_\mathrm{g}$ as the product of the growth deformation $\mathbf{F}_\mathrm{g}$ and the elastic deformation $\mathbf{F}_\mathrm{e}$ (*Figure 2—figure supplement 4*) (*Rodriguez et al., 1994*). We used $\mathbf{F}_\mathrm{g} = \begin{pmatrix} 1+g & 0 \\ 0 & 1 \end{pmatrix}$ for the biofilm and residual layers to describe their 1D growth ($g > 0$) in the $X_1$ direction, and we set $\mathbf{F}_\mathrm{g}$ to be the identity matrix $\mathbf{I}$ for the non-growing agar substrate. The growth-induced compressive strain is thus $\varepsilon = g/(1+g)$. To account for the nonlinear stress-strain behavior of materials undergoing large deformations, all three layers were modeled as neo-Hookean materials. The strain energy density of each layer is given by *Ogden (1997)*:

$$\Psi(\mathbf{F}_\mathrm{e}) = \frac{\mu_\mathrm{e}}{2}(I_\mathrm{C} - 2 - 2\ln J) + \frac{\lambda_\mathrm{e}}{2}(\ln J)^2,$$

where $\mu_\mathrm{e}$ and $\lambda_\mathrm{e}$ are the Lamé parameters, and they are related to the shear modulus $G$ and Poisson's ratio $\nu$ by

$$\mu_\mathrm{e} = G, \ \ \lambda_\mathrm{e} = \frac{2G\upsilon}{1 - 2\upsilon}.$$

$I_\mathrm{C} = \mathrm{tr}(\mathbf{F}_\mathrm{e}^\mathrm{T}\mathbf{F}_\mathrm{e})$ is the first invariant of the right Cauchy-Green deformation tensor $\mathbf{C} = \mathbf{F}_\mathrm{e}^\mathrm{T}\mathbf{F}_\mathrm{e}$, and $J = \det(\mathbf{F}_\mathrm{e})$. The total elastic energy of the system can thus be calculated by

$$\Pi = \int_{\Omega_\mathrm{f}} \Psi\left(\mathbf{F}_{\mathrm{e},\,\mathrm{f}}\right) J_{\mathrm{g},\mathrm{f}} dX + \int_{\Omega_\mathrm{r}} \Psi\left(\mathbf{F}_{\mathrm{e},\,\mathrm{r}}\right) J_{\mathrm{g},\mathrm{r}} dX + \int_{\Omega_\mathrm{s}} \Psi\left(\mathbf{F}_{\mathrm{e},\,\mathrm{s}}\right) J_{\mathrm{g},\mathrm{s}} dX ,$$

where $\Omega_\mathrm{f/r/s}$ denotes the volume occupied by biofilm/residual/substrate in the initial undeformed

reference configuration, and $J_g$ = det($F_g$) specifies the volume element change following growth. We assumed that the present instability pattern always seeks the lowest potential energy among all possible configurations at any time during biofilm growth, neglecting the viscoelasticity and plasticity of the biomaterials that could potentially lead to hysteresis in mechanical instability.

## Finite element simulations

For the computational model, we considered a rectangular domain $\Omega = \Omega_f \bigcup \Omega_r \bigcup \Omega_s$ = [0, $L$]×[0, $h_f$+$h_r$+$h_s$] composed of three layers, where $L$ denotes the size of the system. We use subscripts 1 and 2 to denote the horizontal and vertical components, respectively. Numerically, the task is to calculate the displacement field $u_i$ = $x_i$ - $X_i$ that minimizes the total potential energy, that is $u = \arg\min_{u \in V_u}\Pi$, where $V_u$ is the function space that satisfies the boundary conditions on $u$. Without loss of generality, we considered a scenario in which the biofilm and residual layers grow together but are confined by the left and right walls of the bottom fixed domain $\Omega$, that is, the boundary conditions were set by $u_1|_{X_1=0} = u_1|_{X_1=L} = u_2|_{X_2=0} = 0$ (*Figure 2—figure supplement 4*). The nonlinear constrained minimization problem was implemented in the open-source computing platform FEniCS (*Alnæs et al., 2015*). The computational model was discretized by first-order triangular elements generated by Gmsh (*Geuzaine and Remacle, 2009*), and the accuracy of the results was verified by mesh refinements. A growth increment of $\Delta g$ = 0.002 was employed in the simulations, up to a maximum of 1. For each step, we computed the equilibrium configuration $x$ and the Green-Lagrange strain tensor $e$ = 0.5($F_e^T F_e$ – $I$) of the system. The critical condition for wrinkling instability was identified as a vertical displacement of the biofilm that surpassed the threshold value (0.01$h_f$). We further calculated the deviatoric strain tensor $e'_{ij}$ = $e_{ij}$ – 0.5$\delta_{ij}e_{kk}$ and the von Mises equivalent strain $\varepsilon_{vM}$ = $(2e'_{ij} e'_{ij} /3)^{1/2}$ (*Jones, 2009*) to visualize the strain distribution among the three layers. All results were visualized by Paraview software (*Ahrens et al., 2005*). For the model parameters, we set $h_r/h_f$ = 0.3 based on the measured thickness values from experiments, and $h_s/h_f$ = 10 to represent the thick substrate. The stiffness contrast $G_r/G_f$ = 0.1 was used according to the optimal fitting value from theoretical curves, and we varied $G_f/G_s$ from 0.02 to 10 to correspond to the experimental conditions. In all simulations, $L$ was set to be larger than 10 times the wavelength to minimize the finite size effect, and the Poisson's ratios of all three layers were set to be 0.45 to ensure convergence of the algorithm.

## Statistical methods

Error bars correspond to standard deviations of the means. Standard *t*-tests were used to compare treatment groups and are indicated in each figure legend. Tests were always two-tailed and unpaired/paired as demanded by the details of the experimental design. All statistical analyses were performed using GraphPad Prism software.

## Software availability

The custom-written MATLAB scripts and simulation codes used in this study are available at https://github.com/f-chenyi/biofilm-morphogenesis (*Fei, 2019*; copy archived at https://github.com/elifesciences-publications/biofilm-morphogenesis).

## Acknowledgements

This work was supported by the Howard Hughes Medical Institute (BLB), National Science Foundation Grants MCB-1713731 (BLB) and MCB-1344191 (to BLB, HAS, and NSW), NIH Grant 2R37GM065859 (BLB), the NSF through the Princeton University Materials Research Science and Engineering Center DMR-1420541, and the Max Planck Society-Alexander von Humboldt Foundation (BLB). JY holds a Career Award at the Scientific Interface from the Burroughs Wellcome Fund. We thank Dr. Qiuting Zhang and Dr. Jie Yin for helpful discussions; Dr. Paul Shao, Dr. Yao-Wen Yeh, Prof. Craig Arnold, and Keyence Corporation for support in optical profiling; Dr. Antonio Perazzo for help in rheological measurements; and Dr. Jindong Zan, Dr. Donald A Winkelmann, and Dr. John Schreiber for assistance with SEM sample preparation.

# Additional information

## Funding

| Funder | Grant reference number | Author |
|---|---|---|
| Burroughs Wellcome Fund | Career Award at the Scientific Interface 1015763 | Jing Yan |
| National Science Foundation | MCB-1344191 | Ned S Wingreen<br>Howard A Stone<br>Bonnie L Bassler |
| National Science Foundation | DMR-1420541 | Andrej Košmrlj<br>Howard A Stone |
| Howard Hughes Medical Institute | | Bonnie L Bassler |
| National Institutes of Health | 2R37GM065859 | Bonnie L Bassler |
| Max Planck Society-Alexander von Humboldt Foundation | | Bonnie L Bassler |
| National Science Foundation | MCB-1713731 | Bonnie L Bassler |

The funders had no role in study design, data collection and interpretation, or the decision to submit the work for publication.

## Author contributions

Jing Yan, Conceptualization, Resources, Data curation, Formal analysis, Funding acquisition, Validation, Investigation, Visualization, Methodology, Writing—original draft, Writing—review and editing; Chenyi Fei, Conceptualization, Resources, Data curation, Software, Formal analysis, Validation, Investigation, Visualization, Methodology, Writing—original draft, Writing—review and editing; Sheng Mao, Software, Formal analysis, Investigation, Methodology, Writing—review and editing; Alexis Moreau, Formal analysis, Investigation, Methodology; Ned S Wingreen, Formal analysis, Supervision, Writing—review and editing; Andrej Košmrlj, Software, Formal analysis, Supervision, Methodology, Writing—review and editing; Howard A Stone, Conceptualization, Formal analysis, Supervision, Funding acquisition, Investigation, Methodology, Writing—original draft, Project administration, Writing—review and editing; Bonnie L Bassler, Conceptualization, Formal analysis, Supervision, Funding acquisition, Validation, Investigation, Writing—original draft, Project administration, Writing—review and editing

## Author ORCIDs

Jing Yan https://orcid.org/0000-0003-2773-0348
Chenyi Fei http://orcid.org/0000-0002-8287-4347
Sheng Mao http://orcid.org/0000-0001-9468-5095
Ned S Wingreen http://orcid.org/0000-0001-7384-2821
Andrej Košmrlj http://orcid.org/0000-0001-6137-9200
Howard A Stone http://orcid.org/0000-0002-9670-0639
Bonnie L Bassler http://orcid.org/0000-0002-0043-746X

## Decision letter and Author response

Decision letter https://doi.org/10.7554/eLife.43920.047
Author response https://doi.org/10.7554/eLife.43920.048

# Additional files

## Supplementary files
• Source data 1. Raw data for *Supplementary file 1* – Table S1.
DOI: https://doi.org/10.7554/eLife.43920.043

• Supplementary file 1. Summary of biomaterial parameters for *V. cholerae* biofilms. Table S1 reports the measured biomaterial parameters for *V. cholerae* biofilms grown on different concentrations of agar substrates. These measurements include the shear modulus of the substrate and the biofilm, the thickness of the biofilm and the residual layer, and the wavelength of the biofilm surface pattern.
DOI: https://doi.org/10.7554/eLife.43920.044

• Transparent reporting form
DOI: https://doi.org/10.7554/eLife.43920.045

## Data availability

All data generated or analyzed during this study are included in the manuscript and supporting files. Source data files have been provided for all figures, tables and figure supplements.

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
