## [Decision Letter]

Thank you for submitting your article "Mechanical instability and interfacial energy drive biofilm morphogenesis" for consideration by *eLife*. Your article has been reviewed by three peer reviewers, and the evaluation has been overseen by a Reviewing Editor and Naama Barkai as the Senior Editor. The following individuals involved in review of your submission have agreed to reveal their identity: Brian Hammer (Reviewer #3).

The reviewers have discussed the reviews with one another and the Reviewing Editor has drafted this decision to help you prepare a revised submission.

This paper presents an interesting analysis of the mechanical origins of wrinkling in bacterial biofilms. There are several conclusions drawn from the combined modeling and experimental analysis: 1) a role for delamination and the subsequent dynamics of wrinkles in early and late periods of biofilm development, 2) the presence of a third layer that is important for recapitulating behavior on different surfaces, 3) a key role of interfacial energies in biofilm development and how they can be altered in mutants, 4) changes to the shape of the biofilm and how they are related to wrinkling. Overall, the reviewers thought the paper was well written and relatively clear. They did, however, identify several points (see below) that would need to be addressed in a revision.

1) Several aspects of how interfacial free energy shapes a biofilm were not fully convincing and need additional explanation and, in some cases, additional experiments.

a) What does "adhere" mean (subsection “Interfacial energy controls blister development dynamics and interactions between blisters”, second paragraph)? The fluorescence image in Figure 4C seems to suggest that there is a space at the inner face of blisters. This suggests that there can still be the liquid-biofilm interface at the "adhered" inner surfaces – is this the case? Additional experiments are needed to clarify the nature of the "adhered" inner surfaces. In particular, the reviewers felt that electron microscopy would be critical to addressing how biofilm layers "adhere". Alternatively, it was suggested that one could perform the fluorescent-beads experiment by Wilking et al. (https://www.ncbi.nlm.nih.gov/pubmed/23271809). If biofilm layers are indeed adhered to reduce the interfacial energy, the fluorescence beads should be absent at the high regions of blisters.

b) The conclusion that buckling results from a mechanical instability indicates that the transition from flat to buckling should happen virtually instantaneously. This should be demonstrated.

c) For the positions of blisters, particularly in the regime where they are relatively spread out, the stress relief hypothesis should mandate that there is an anticorrelation between their locations (i.e. blistering in one place should prevent nearby blisters). Is this the case?

d) The conclusion of their last section, that positions of the mechanical instabilities relate to contour evolution, makes the strong prediction that if they generate blisters in predefined locations (as in Figure 3E), they should be able to control contour shape.

e) This study is relying on an assumption that the mechanical property of biofilm is spatially homogeneous, and the compressive stress is uniformly applied. However, experimental support for these basic assumptions is lacking. The authors reported that the amplitudes of wrinkles and blisters are heterogeneous in some conditions. What is the source of the heterogeneity?

f) The authors introduced two models describing biofilm morphology development (Introduction, second paragraph). However, these two models are not exclusive. In fact, the authors described the coupling between gene regulation and mechanical instability later in the text (Discussion, end of second paragraph), and this conclusion can be reached by existing literature. This aspect of the manuscript needs clarification and more attention to precision of language.

2) What is the nature of the debris layer? The prediction of the "debris" layer arising from discrepancies between a simple bilayer model and the experiment results was intriguing, but the chemical characteristics of the debris layer are not well described. First, the cross sectional images depicted in Figure 3C and 5A are difficult to interpret and this presentation could be improved. Second, the suggestion that the debris layer between a biofilm and agar matrix lacks cells is supported by the lack of *mKate2* expression, but the absence of signal may also result from poor expression of the reporter at this interface.

3) What is the role of growth and cell death? The model does an excellent job of explaining the behavior of the blistering, particularly once they incorporate the third, soft layer between the biofilm and the substrate. However, a major input into the model are the locations at which growth is taking place, which is the source of the buildup of stress. It was unclear to me what was being assumed about growth – is it distributed throughout the colony, or only at the edge? Along those lines, have the authors tried to measure the growth directly using fluorescence as a proxy? Finally, cell death could also be an important factor; as the authors note, Asally et al., 2012, suggest that cell death relieves mechanical stress build-up. Is the blistering that the authors observe in *V. cholerae* independent of this effect, or do they also see increased cell death preceding locations at which blistering occurs?

4) The connections to eukaryotic biology/tissue development were uncertain and overly speculative. The concepts regarding mechanical properties and processes studied in this work are not new in the research field of eukaryotic tissue development. For example, the authors claim that interfacial energy was not investigated in eukaryotic morphogens, and thus, "exploiting interfacial energy differences to dictate morphology could be a unique feature to bacterial communities". This statement is confusing because there is an extensive body of work studying the contributions of interfacial energy in the context of tissue development (e.g. Differential Adhesion Hypothesis, doi:10.1115/1.1449491, 10.1016/j.ydbio.2004.11.012). Cell sorting by differential interfacial energy is also a big topic of research in eukaryotic developmental biology. The authors should tone down the claims of novelty throughout the manuscript and not 'oversell' the connections to eukaryotic biology.

---

## [Author Response]

1) Several aspects of how interfacial free energy shapes a biofilm were not fully convincing and need additional explanation and, in some cases, additional experiments.

As detailed below, in the revised manuscript, we provide significant new data to underpin our claims for how interfacial free energy determines biofilm morphogenesis, and, in particular, the morphology and internal architecture of individual blisters.

a) What does "adhere" mean (subsection “Interfacial energy controls blister development dynamics and interactions between blisters”, second paragraph)? The fluorescence image in Figure 4C seems to suggest that there is a space at the inner face of blisters. This suggests that there can still be the liquid-biofilm interface at the "adhered" inner surfaces – is this the case? Additional experiments are needed to clarify the nature of the "adhered" inner surfaces. In particular, the reviewers felt that electron microscopy would be critical to addressing how biofilm layers "adhere". Alternatively, it was suggested that one could perform the fluorescent-beads experiment by Wilking et al. (https://www.ncbi.nlm.nih.gov/pubmed/23271809). If biofilm layers are indeed adhered to reduce the interfacial energy, the fluorescence beads should be absent at the high regions of blisters.

We thank the reviewers for bringing up this important point. The appearance of a gap between the two sides of the blister in Figure 4C is due to a layer of dead cells located underneath the living, fluorescent layer. We now describe this layer in the text. We show that the living and dead cell layers can be distinguished by SytoX Green staining (see new results in Figure 4—figure supplement 2B). In the new sideview image, when a blister forms, it is evident that the dead cell layer adheres to the live cell layer leaving an empty area underneath the blister. In the manuscript, we do not consider the dead cell layer separately from the live cell layer in the mechanical instability model for two reasons: 1) the dead cell layer is always attached to the live cell layer; 2) when we measured the biofilm moduli, the dead and live cell layers were removed together from the substrate and measured together. Therefore, for the purpose of the mechanical instability analysis, we treat the live and dead cell layers together as one “biofilm layer”.

We suspect that the dead cell layer forms due to limited oxygen penetration into the biofilm. Indeed, lack of oxygen availability in biofilms is commonly observed (see, for example, Bellin et al. Nat. Commun. 7, 10535).

We provide evidence that the sides of blisters are in contact with each other, which is not obvious from the fluorescence images. We have now performed SEM imaging as suggested by the reviewers (new Figure 4—figure supplement 2A). The SEM image shows that the two sides of a blister are indeed contacting one another. While dehydration occurs during SEM sample preparation and gives rise to a honeycomb-like structure (see, for example, O’Brian et al. Biomaterials. 25, 1077-1086), the tight interface between the two sides as well as the gap underneath the blister are clearly visible.

b) The conclusion that buckling results from a mechanical instability indicates that the transition from flat to buckling should happen virtually instantaneously. This should be demonstrated.

We thank the reviewers for this creative suggestion. We have now performed a time course and we provide the results in the new Figure 1—figure supplement 2. We find that between 31-33 h, the transition from flat to wrinkled biofilm morphology takes place, indeed, within a narrow time window. The transition is not, however, instantaneous. There are two reasons for this: 1) the accumulation of strain, which is due to cell growth, is gradual; 2) the wrinkling instability is a second order transition so there can be no sudden jump in wrinkle amplitude as in first order phase transitions.

c) For the positions of blisters, particularly in the regime where they are relatively spread out, the stress relief hypothesis should mandate that there is an anticorrelation between their locations (i.e. blistering in one place should prevent nearby blisters). Is this the case?

Yes! Indeed, this is what we observe. As shown in Figure 4B, when an individual blister emerges, a consequence is that the nearby wrinkles flatten out (see the contour line around the blister). In the revised manuscript, we have included a discussion of this feature of the morphology development.

d) The conclusion of their last section, that positions of the mechanical instabilities relate to contour evolution, makes the strong prediction that if they generate blisters in predefined locations (as in Figure 3E), they should be able to control contour shape.

Indeed, this is the case. Figure 3E shows the biofilm at 48 h, a timepoint at which the undulation in the biofilm contour is visible but not yet particularly significant. In response to this comment, we have now added new Figure 6—figure supplement 2A showing another biofilm with a predefined blister position. This biofilm was grown for 60 h, so it exhibits a more pronounced contour undulation. The new figure shows that the initial surface imperfections not only determine the positions of the blisters but also the overall shape of the biofilm contour.

e) This study is relying on an assumption that the mechanical property of biofilm is spatially homogeneous, and the compressive stress is uniformly applied. However, experimental support for these basic assumptions is lacking. The authors reported that the amplitudes of wrinkles and blisters are heterogeneous in some conditions. What is the source of the heterogeneity?

The mechanical properties of the biofilm that we have quantified rely on bulk rheological measurements. Bulk measurements do not, unfortunately, enable us to interrogate sources of mechanical heterogeneity in growing biofilms. Future experiments mapping the spatial distribution of biofilm mechanics, for example through AFM measurements, are required to properly address this point. While fascinating, such measurements are beyond the scope of the current study.

On the other hand, we agree that some of the morphological heterogeneity (i.e., wavelength fluctuation) might arise from modest variations in biofilm material properties. However, the length scales of fluctuations in biofilm microstructures (~5 μm, see Figure 5B in Yan et al., 2018) are much smaller than the typical morphological wavelength (~400 μm). Thus, it is reasonable to assume that the averaged material properties on a continuum level are approximately constant. Additionally, prior to wrinkling/delamination, a growing biofilm displays no obvious heterogeneity in the flat region at the edge as judged by bright field and transmission imaging, which suggest homogeneous material properties.

Regarding the reviewer’s concern about the homogeneity of compressive stress, the current theoretical and computational models were developed to help rationalize the wavelength changes under various experimental conditions. The wavelengths of wrinkles due to mechanical instability are not sensitive to the stress distribution, but are determined by the thickness of the biofilm and relevant material properties. Thus, for simplicity, we assumed 1D and uniform compression in our theory and simulation. In the experiments, we do agree that there could be nonuniform compressive stress. We expect that nonuniform compressive stress would affect the locations at which wrinkles first appear, and moreover, make the wrinkle amplitudes heterogenous (see, for example, new Figure 2—figure supplement 2).

We do know that heterogeneity in blister size results from random events that dictate when and where a blister will emerge. In the text, we have offered the hypothesis that blister positions are determined by microscopic surface imperfections. Blisters will emerge at different times and at different places in the growing biofilms depending on when a surface defect is encountered during biofilm expansion (once enough mechanical strain is accumulated). The different ages of blisters naturally lead to their heterogeneous heights.

*f) The authors introduced two models describing biofilm morphology development (Introduction, second paragraph). However, these two models are not exclusive. In fact, the authors described the coupling between gene regulation and mechanical instability later in the text (Discussion, end of second paragraph), and this conclusion can be reached by existing literature. This aspect of the manuscript needs clarification and more attention to precision of language.*

We thank the reviewers for pointing out this issue. In the updated manuscript, we have clarified the distinguishing features of the two models, and we lay out how they differ from our current mechanomorphogenesis model.

2) What is the nature of the debris layer? The prediction of the "debris" layer arising from discrepancies between a simple bilayer model and the experiment results was intriguing, but the chemical characteristics of the debris layer are not well described. First, the cross sectional images depicted in Figure 3C and 5A are difficult to interpret and this presentation could be improved. Second, the suggestion that the debris layer between a biofilm and agar matrix lacks cells is supported by the lack of mKate2 expression, but the absence of signal may also result from poor expression of the reporter at this interface.

We appreciate this question as it caused us to focus very hard on this important point and we have significantly revised the text to address this issue. The presence of an intermediate layer is critical to our interpretation of the scaling relationship between the substrate stiffness and the biofilm wrinkle wavelength. While the exact chemical nature of this intermediate layer is difficult to fully analyze and define, in the revised manuscript, in addition to the data provided in Figure 2—figure supplement 2, we now provide two new pieces of evidence to show that the layer is composed primarily of biofilm matrix, most notably polysaccharide. First, CFU measurements in new Figure 2—figure supplement 3A show that there are significantly fewer live cells in this layer compared to the bulk biofilm. Second, new Figure 2—figure supplement 3B demonstrates that India ink, a common counter stain used for the detection of polysaccharides (see for example Qadri et al. Infect. Immun. 73, 6577), cannot penetrate this residual layer showing that this layer consists primarily of polysaccharide. We do not exclude the possibility that the residual layer also contains dead cells, proteins, lipids, etc. In light of the new experiments and our deeper understanding of this layer, we think that it is more appropriate to call this layer a “residual” layer, rather than a “debris” layer as the latter has connotations that we do not mean to imply. All of the text, figures, and formulas have been updated to reflect this change.

3) What is the role of growth and cell death? The model does an excellent job of explaining the behavior of the blistering, particularly once they incorporate the third, soft layer between the biofilm and the substrate. However, a major input into the model are the locations at which growth is taking place, which is the source of the buildup of stress. It was unclear to me what was being assumed about growth – is it distributed throughout the colony, or only at the edge? Along those lines, have the authors tried to measure the growth directly using fluorescence as a proxy? Finally, cell death could also be an important factor; as the authors note, Asally et al., 2012, suggest that cell death relieves mechanical stress build-up. Is the blistering that the authors observe in *V. cholerae* independent of this effect, or do they also see increased cell death preceding locations at which blistering occurs?

We thank the reviewers for bringing up this central point. Indeed, in the original version, we had, as is common in the literature, assumed that growth occurs primarily at the edge of the biofilm due to nutrient limitation in the center of the biofilm. However, we had not provided any proof. To demonstrate that this is indeed the case, in the new Figure 4—figure supplement 2B, we use SytoX Green dead-cell staining to show that there is a thin (~1 mm) annulus at the edge of the biofilm in which significantly reduced cell death occurs relative the core of the biofilm, supporting the hypothesis that primary growth occurs at the biofilm edge. In contrast to what Asally et al., 2012, reported, under our conditions (see for instance Video 3), the instability pattern can emerge from the edge of the biofilm where cell death is minimal.

Regarding the position of the blister, we do not observe the local concentration of dead cells prior to blister formation (see new Figure 1 – —figure supplement 2). Therefore, we conclude that the mechanism proposed by Asally et al. is not applicable to the formation of radial wrinkles/blisters in the current system. However, we do suggest Asally’s mechanism for local cell death could lead to the labyrinthine pattern at the center of the biofilm (see Figure 1, for example). Indeed, Figure 2—figure supplement 1C shows our quantitation of the wavelength at the center of the biofilm and that it is different from that at the edge. Thus, the center region is likely following Asally et al.’s model, while the edge is not.

4) The connections to eukaryotic biology/tissue development were uncertain and overly speculative. The concepts regarding mechanical properties and processes studied in this work are not new in the research field of eukaryotic tissue development. For example, the authors claim that interfacial energy was not investigated in eukaryotic morphogens, and thus, "exploiting interfacial energy differences to dictate morphology could be a unique feature to bacterial communities". This statement is confusing because there is an extensive body of work studying the contributions of interfacial energy in the context of tissue development (e.g. Differential Adhesion Hypothesis, doi:10.1115/1.1449491, 10.1016/j.ydbio.2004.11.012). Cell sorting by differential interfacial energy is also a big topic of research in eukaryotic developmental biology. The authors should tone down the claims of novelty throughout the manuscript and not 'oversell' the connections to eukaryotic biology.

As suggested, we have now toned down the amount of speculation about eukaryotic biology in the text and we removed the specific examples describing how mechanical instabilities shape eukaryotic development. We also thank the reviewers for directing us to the valuable literature on how interfacial tension drives eukaryotic tissue development; we had not been aware of these papers. After careful reading, we now think our main contribution is to reveal and quantify the involvement of interfacial energy in the context of mechanical instability during morphology development. We have updated the text accordingly.